# Position: Graph Matching Systems Deserve Better Benchmarks

**Indradyumna Roy** [1]  **Saswat Meher** [1]  **Eeshaan Jain** [2]  **Soumen Chakrabarti** [1]  **Abir De** [1]

## Abstract

Data sets used in recent work on graph similarity scoring and matching tasks suffer from significant limitations. Using Graph Edit Distance (GED) as a showcase, we highlight pervasive issues such as train-test leakage and poor generalization, which have misguided the community's understanding and assessment of the capabilities of a method or model. These limitations arise, in part, because preparing labeled data is computationally expensive for combinatorial graph problems. We establish some key properties of GED that enable scalable data augmentation for training, and adversarial test set generation. Together, our analysis, experiments and insights establish new, sound guidelines for designing and evaluating future neural networks, and suggest open challenges for future research.

## 1. Introduction

Graph matching systems are critical for aligning structured objects and relationships, underpinning applications such as scene graph analysis (Chen et al., 2020; van Engelenburg et al., 2023), molecular compound comparison (Garcia-Hernandez et al., 2019; Rica et al., 2021), knowledge graphs (Tang et al., 2020; Fang et al., 2023), social network analysis (Donnat & Holmes, 2018) and fraud detection in e-commerce (Nguyen et al., 2023; Yang & Cogill, 2013; Calabuig et al., 2023). Accurate graph similarity assessment is key to deriving actionable insights across these domains. The Graph Edit Distance (GED) framework unifies diverse similarity notions, both symmetric and asymmetric, through its variable cost setup. GED calculates the minimum-cost sequence of edit operations—including node and edge additions, deletions, and substitutions—to transform one graph into another. This *variable-cost* formulation allows GED

to act as a generalized framework for graph matching, subsuming various classic problems such as subgraph isomorphism (Messmer & Bunke, 1994; 1998), maximum common subgraph (MCS) (Bunke, 1997; Brun et al., 2012), and equal-cost GED as special cases.

One major obstacle to adopting combinatorial GED-based graph matching is its computational cost, stemming from the NP-hardness of GED, which becomes infeasible for graphs with more than a few dozen nodes (Blumenthal & Gamper, 2020). Neural approaches are envisioned as promising alternatives, ideally offering two key advantages: (A) In retrieval setups, where a query graph is matched against a large corpus, neural networks can provide fast, approximate solutions to the quadratic assignment problem (QAP) by leveraging hardware accelerators, provided they generalize to unseen graphs drawn from the same distribution as the training data. (B) For larger graphs, neural networks trained on smaller graphs must generalize to approximate GED effectively during inference. Both use cases depend on the neural models' ability to generalize beyond the training distribution, ensuring scalability and robust performance for real-world applications.

Recent neural approaches for GED estimation typically learn from datasets of graph pairs annotated with ground truth GED values. These methods broadly fall into two categories: (A) coarse graph-level fixed-size representations that approximate GED using embedding space similarity, prioritizing scalability over accuracy, and (B) fine-grained interaction models that represent graphs as *sets* of embeddings, enabling richer interaction modeling but at higher computational cost.

Despite their promise, these approaches suffer from three critical issues. **First**, most studies (Bai et al., 2019a; Piao et al., 2023; Ranjan et al., 2022; Qin et al., 2021; Zhuo & Tan, 2022) rely on a small collection of datasets reused over the past five years. These datasets suffer from critical train-test leakage, undermining evaluations of model generalization and effectiveness. This has led to recent skepticism regarding the utility of embedding-based methods for GED estimation, with some studies favoring unsupervised approaches (Gao et al., 2021). However, we argue that such conclusions are premature, as leakage issues confound evaluations and can lead to incorrect judgments

[1]IIT Bombay, Mumbai, India [2]EPFL, Lausanne, Switzerland. Correspondence to: Indradyumna Roy <indraroy15@cse.iitb.ac.in>, Abir De <abir@cse.iitb.ac.in>.

*Proceedings of the 42nd International Conference on Machine Learning*, Vancouver, Canada. PMLR 267, 2025. Copyright 2025 by the author(s).

about model performance. **Second**, simple models based on comparing fixed-size whole-graph representations have occasionally been observed to outperform more complex interaction-based models (Ranjan et al., 2022), going against conventional wisdom regarding trade-offs between representational granularity and expressive power. In view of train-test leakage, it becomes critical to carefully check if the discrepancy should be attributed to leakage or some other phenomenon. **Third**, most neural models fail to align with principles underlying traditional GED computation, such as incorporating variable edit costs, which are crucial to GED's universality and adaptability. Algorithmically aligned neural architectures are known to generalize better (Xu et al., 2019b); thus, methods that capture the full generality of GED are likely to be more effective. The lack of robust benchmarks makes it challenging to pinpoint specific limitations of current models and determine whether reported performance gains reflect genuine advances or artifacts of flawed evaluation protocols.

In this paper, we establish the severity of issues in the current development of graph matching systems, which stem primarily from the lack of reliable benchmarks. Our contributions are as follows:

**Data Leakage Analysis.** We identify pervasive train-test leakage that has gone unnoticed for years. Specifically, we show that test graph pairs being isomorphic to training graph pairs constitutes leakage, as most models rely on node-order invariant graph neural networks. We quantify the extent of this leakage and identify prior architectures likely impacted by it.

**Deeper Understanding of Variable Cost GED.** We investigate the principles governing optimal edit paths and edit costs in variable cost GED. Under certain assumptions, we propose methods to efficiently generate high-quality training data at scale.

**Adversarial Test Set Design.** Leveraging our understanding of GED, we propose methods to design adversarial test sets that rigorously evaluate whether models can identify the correct ground truth permutations, thereby addressing gaps in current evaluation protocols.

Our work aims to provide actionable insights for improving the design, training, and evaluation of neural models for graph matching.

## 2. Preliminaries

The inputs to GED are the source graph $G = (V, E)$ and the target graph $G' = (V', E')$, where $V, V'$ are node sets and $E \subseteq V \times V, E' \subseteq V' \times V'$ are edge sets. Nodes may optionally have labels from a label set $\mathcal{L}$, defined by a label function $\ell_V : V \cup V' \to \mathcal{L}$. Unless otherwise

mentioned, both graphs are padded with artificial nodes such that $|V| = |V'| = N$, with adjacency matrices $\boldsymbol{A}, \boldsymbol{A}' \in \{0, 1\}^{N \times N}$, where $\boldsymbol{A}[i, j] = 1$ if $(i, j) \in E$ and $\boldsymbol{A}'[i, j] = 1$ if $(i, j) \in E'$. To distinguish actual nodes from artificial nodes, we maintain indicator vectors $\mathbf{I}_G, \mathbf{I}_{G'} \in \{0, 1\}^N$, where $\mathbf{I}_G[i] = 1$ if $i$ is an actual node in $G$ and $\mathbf{I}_G[i] = 0$ otherwise; the same applies for $G'$.

The space of permutation matrices is $\Pi_N = \{\boldsymbol{P} \in \{0, 1\}^{N \times N} \mid \boldsymbol{P}\mathbf{1} = \mathbf{1}, \ \boldsymbol{P}^\top\mathbf{1} = \mathbf{1}\}$, where $\mathbf{1}$ is the all-ones vector. Each $\boldsymbol{P} \in \Pi_N$ defines a *node alignment map*, aligning $i \in V$ to $j \in V'$ if $\boldsymbol{P}_{ij} = 1$, and induces a graph edit path—a sequence of edit operations—from $G$ to $G'$, explicitly constructed using Algorithm 1. Each edit operation is associated with a potentially different cost (described next with cost in brackets), allowing flexibility in modeling application-specific requirements:

(1) **Node insertion** ($c_{\text{NA}}$): $\boldsymbol{P}_{ij} = 1$, $\mathbf{I}_G[i] = 0$, and $\mathbf{I}_{G'}[j] = 1$; (2) **Node deletion** ($c_{\text{ND}}$): $\boldsymbol{P}_{ij} = 1$, $\mathbf{I}_G[i] = 1$, and $\mathbf{I}_{G'}[j] = 0$; (3) **Node substitution** ($c_{\text{NS}}$): $\boldsymbol{P}_{ij} = 1$, $\mathbf{I}_G[i] = 1, \mathbf{I}_{G'}[j] = 1$, and $\ell_V(i) \neq \ell_V(j)$; (4) **Edge insertion** ($c_{\text{EA}}$): $\boldsymbol{P}_{ij} = 1$, $\boldsymbol{P}_{kl} = 1$, $\boldsymbol{A}[i, k] = 0$, and $\boldsymbol{A}'[j, l] = 1$; (5) **Edge deletion** ($c_{\text{ED}}$): $\boldsymbol{P}_{ij} = 1$, $\boldsymbol{P}_{kl} = 1$, $\boldsymbol{A}[i, k] = 1$, and $\boldsymbol{A}'[j, l] = 0$.

All edit operation costs are assumed to be non-negative ($c_{\text{NA}}, c_{\text{ND}}, c_{\text{NS}}, c_{\text{EA}}, c_{\text{ED}} \geq 0$). This general formulation, which assigns distinct costs to different edit operations, defines what we refer to as *variable-cost GED*. Given $\boldsymbol{P} \in \Pi_N$, the total cost of the corresponding edit path is the sum of all edit operation costs. The Graph Edit Distance (GED) is the minimum total cost over all permutations:

$$\text{GED}(G, G') = \min_{\boldsymbol{P} \in \Pi_N} \sum_{o \in \mathcal{O}(\boldsymbol{P})} c(o),$$

where $\mathcal{O}(\boldsymbol{P})$ is the set of edit operations induced by $\boldsymbol{P}$, and $c(o)$ is the cost of an operation $o$. This optimization over permutation matrices, subject to edge consistency constraints, formulates GED as a classical Quadratic Assignment Problem (QAP). There may be multiple optimal edit paths achieving the minimum transformation cost. We denote the set of all such optimal edit paths as $\mathcal{E}^*(G, G')$. The set $\mathcal{E}^*(G, G')$ contains graph edit paths corresponding to all node alignment maps $\boldsymbol{P}^* \in \Pi_N$ that minimize the total cost, defining the optimal transformations from $G$ to $G'$.

A special case is the widely used *equal-cost GED* setting, where all edit operations are assigned the same cost (typically 1). In contrast, the more general *variable-cost GED* allows different costs for each operation type, offering greater expressiveness in modeling nuanced graph similarity, including asymmetric relevance.

# 3. Related Work

Our paper is related to (1) combinatorial solvers for GED, (2) neural approaches for GED estimation and (3) optimal transport. In the following, we briefly review them.

## 3.1. Exact and Approximate solvers for GED.

Since its introduction (Sanfeliu & Fu, 1983), GED has been extensively studied due to its versatility and wide applicability (Bunke & Allermann, 1983; Bunke, 1997). Classical approaches for exact GED computation, such as the A* algorithm (Hart et al., 1968; Riesen et al., 2013), rely on exhaustive enumeration of node mappings with heuristics to prune the search space. However, the NP-hard nature of GED (Zeng et al., 2009) renders these methods computationally infeasible for large graphs (Abu-Aisheh et al., 2017), despite improvements through depth-first heuristics (Abu-Aisheh et al., 2015) and parallel computation (Abu-Aisheh et al., 2018). GED has also been reformulated as a Quadratic Assignment Problem (QAP) (Bougleux et al., 2017) , and subsequently, various relaxations and heuristics (Neuhaus et al., 2006; Riesen & Bunke, 2009b; Fankhauser et al., 2011) have been leveraged to enable approximate solutions. Combinatorial algorithms such as Bipartite Matching (Riesen & Bunke, 2009a), which solves a linear assignment problem on node neighborhoods, and Branch/Branch Tight (Blumenthal & Gamper, 2018), which decompose graphs into branches, provide scalable alternatives. Further, Anchor-Aware GED (Chang et al., 2017) refines lower bounds using anchor-based techniques, while IPFP (Bougleux et al., 2017) jointly optimizes node and edge assignments through QAP. Finally, F2 (Lerouge et al., 2017), based on binary linear programming, provides high-quality lower bounds and is often used for generating ground truth GED values. These methods strike a balance between computational efficiency and accuracy, making them practical for large graphs.

We explored two libraries, GEDLIB (Blumenthal et al., 2019) and NetworkX (Hagberg & Conway, 2020), for GED calculation using combinatorial approaches. GEDLIB allows variable edit cost settings as input and computes both the GED value and the node mapping between two graphs using various combinatorial methods. This node mapping can be utilized to construct graph edit paths. However, GEDLIB provides only a single GED node mapping, whereas NetworkX can generate all possible node mappings corresponding to the GED value while accepting variable edit costs.

## 3.2. Neural Approaches for GED Estimation.

Recent advancements in GED approximation have transitioned from combinatorial heuristics to neural models that exploit distributional characteristics for faster inference with lower amortized costs. Neural models built on Graph Neural Network (GNN) backbones tackle the problem at graph- or node-level granularities. Graph-level methods, such as GMN-Embed (Li et al., 2019), GREED (Ranjan et al., 2022), and GMN-Match (Li et al., 2019), map entire graphs to embeddings, differing in their use of early or late interactions. Node-level approaches, including SimGNN (Bai et al., 2019a), GraphSim (Bai et al., 2020), GOTSim (Doan et al., 2021), and GRAPHEDX (Jain et al., 2024), focus on learning alignments from node-local graph structures.

Several pairwise interaction modules have also been introduced to improve GED estimation. SimGNN (Bai et al., 2019a) proposed a Neural Tensor Network module for computing whole-graph embeddings, which has since been adapted in ERIC (Zhuo & Tan, 2022) and EGSC (Qin et al., 2021). ISONET (Roy et al., 2022) leverages Gumbel-Sinkhorn-based edge alignment, while GRAPHEDX (Jain et al., 2024) integrates node and edge alignments derived from the Kronecker product of node alignments. ERIC (Zhuo & Tan, 2022) combines Siamese architectures with a regularizer to eliminate the need for explicit alignments. H2MN (Zhang et al., 2021) models higher-order node interactions using hypergraph convolutions, and EGSC (Qin et al., 2021) refines graph embeddings through embedding fusion networks. Collectively, these neural approaches leverage end-to-end learning pipelines, offering significant advancements in GED approximation. However, they face challenges such as dependence on uniform-cost GED datasets, limited scalability to large graphs, and an inability to directly infer interpretable edit paths essential for practical use.

While comparing existing neural models, GRAPHEDX stands out as the only model inherently designed to accommodate variable edit costs. For other models, variable costs can be integrated by encoding them as node features. With respect to graph edit path generation, GRAPHEDX, ISONET, and GEDGNN can produce graph edit paths. However, GEDGNN requires fine-grained supervision during training, whereas GRAPHEDX and ISONET do not.

## 3.3. Optimal Transport

Optimal Transport (OT) has emerged as a cornerstone in machine learning for aligning both discrete and continuous distributions, primarily due to the entropic regularization framework (Cuturi, 2013), which leverages the differentiable Sinkhorn network for scalable computation on modern hardware. Similar to combinatorial heuristics that reformulate the QAP as a Linear Sum Assignment Problem (LSAP) on richer subgraph structures (Serratosa & Cortés, 2015; Gaüzère et al., 2014; Carletti et al., 2015), neural GED models utilize GNNs to learn sets of local structural

**Algorithm 1** Construct Edit Path from Permutation Matrix

---

**Require:** Source graph $G = (V, E, \ell_V)$, target graph $G' = (V', E', \ell_V)$, permutation matrix $\boldsymbol{P} \in \Pi_N$, dummy indicators $\mathbf{I}_G, \mathbf{I}_{G'}$.
**Ensure:** Edit path $\mathcal{O}(\boldsymbol{P})$ from $G$ to $G'$.
 1: Initialize $\mathcal{O}(\boldsymbol{P}) \leftarrow \emptyset$
 2: Compute $\widehat{\boldsymbol{A}'} \leftarrow \boldsymbol{P}\boldsymbol{A}'\boldsymbol{P}^\top$
 3: ▷ **Process edge deletions.**
 4: **for all** $(i, k) \in [N] \times [N]$ **do**
 5:     **if** $\boldsymbol{A}[i, k] = 1$ and $\widehat{\boldsymbol{A}'}[i, k] = 0$ **then**
 6:        Add **Edge Deletion** $(i, k)$ to $\mathcal{O}(\boldsymbol{P})$
 7: ▷ **Process node edit operations.**
 8: **for** $i = 1$ to $N$ **do**
 9:     Let $j$ satisfy $\boldsymbol{P}_{ij} = 1$.
10:     **if** $\mathbf{I}_G[i] = 1$ and $\mathbf{I}_{G'}[j] = 0$ **then**
11:        Add **Node Deletion** $(i)$ to $\mathcal{O}(\boldsymbol{P})$
12:     **if** $\mathbf{I}_G[i] = 0$ and $\mathbf{I}_{G'}[j] = 1$ **then**
13:        Add **Node Insertion** $(i)$ to $\mathcal{O}(\boldsymbol{P})$
14:     **if** $\mathbf{I}_G[i] = 1$ and $\mathbf{I}_{G'}[j] = 1$ and $\ell_V(i) \neq \ell_V(j)$ **then**
15:        Add **Node Substitution** $(i \to j)$ to $\mathcal{O}(\boldsymbol{P})$
16: ▷ **Process edge insertions.**
17: **for all** $(i, k) \in [N] \times [N]$ **do**
18:     **if** $\boldsymbol{A}[i, k] = 0$ and $\widehat{\boldsymbol{A}'}[i, k] = 1$ **then**
19:        Add **Edge Insertion** $(i, k)$ to $\mathcal{O}(\boldsymbol{P})$
20: **return** $\mathcal{O}(\boldsymbol{P})$

---

embeddings for each graph and apply OT between these embedding sets to approximate GED as the Wasserstein distance. This approach involves aligning two discrete distributions with uniform mass, and for variable-sized graphs, it extends naturally to unbalanced OT (Pham et al., 2020; Chizat et al., 2018). Some recent methods, such as GOT-Sim (Doan et al., 2021), use combinatorial OT solvers for node mapping, while others like ISONET (Roy et al., 2022) and GRAPHEDX (Jain et al., 2024) leverage Sinkhorn-based differentiable permutation surrogates for efficient approximations. Advancements in OT have expanded its application to aligning distributions across metric spaces through the Gromov-Wasserstein (GW) distance (Mémoli, 2011; Peyré et al., 2016; Xu et al., 2019a), which compares intra-domain dissimilarity matrices and provides a direct parallel to QAP in graph matching under pairwise edge constraints. Fused GW (Vayer et al., 2020; Brogat-Motte et al., 2022) integrates Wasserstein and Gromov-Wasserstein distances, enabling the simultaneous optimization of graph topology and node label alignment for richer and more flexible graph matching frameworks.

## 4. Analysis of variable cost GED

We now examine variable-cost Graph Edit Distance (GED) and the role of node permutations in constructing valid graph

transformations. We show that, under certain conditions, the optimal edit path remains invariant across a broad range of cost settings. This invariance enables scalable data augmentation from a single alignment and sets the stage for adversarial evaluation protocols introduced in Section 5.

### 4.1. Edit Path Ordering: Edge Deletions → Node {Insertions/Deletions/Substitutions} → Edge Insertions

Algorithm 1 generates an interpretable edit path from the permutation matrix $\boldsymbol{P}$, ensuring a valid sequence of operations. The algorithm first performs **edge deletions**, as nodes can only be safely deleted if all incident edges have been removed—deleting a node with existing edges would result in an invalid intermediate state. Next, it processes **node operations**, including deletions, insertions, and substitutions, based on the alignment $\boldsymbol{P}$. Finally, it handles **edge insertions**, since edges involving a newly added node can only be added after the node itself exists. This structured sequence ensures that all intermediate graphs during the transformation remain valid and interpretable, guaranteeing a logical progression from $G$ to $G'$.

This underscores the tight coupling between a permutation matrix $\boldsymbol{P}$ and the induced edit path: once $\boldsymbol{P}$ is fixed, Algorithm 1 deterministically yields a unique, valid sequence of edit operations based solely on the input graphs. At the same time, it reveals a crucial decoupling: the final GED value is determined not by the path itself, but by the cost vector $\mathcal{C}$ applied to this sequence. This separation allows us to formally study how cost configurations influence optimal alignments. In particular, we leverage this distinction in the propositions that follow to characterize when and how the edit path—and consequently, the optimal alignment—remains invariant under different choices of $\mathcal{C}$, opening up key implications for learning and evaluation.

**Proposition 4.1.** *If the substitution cost $c_{NS}$ satisfies $c_{NS} < c_{NA} + c_{ND}$, then no optimal edit path includes both a node addition and a node deletion.*

*Proof.* The GED between $G$ and $G'$ can be computed recursively by minimizing over node pairs $(u, v)$, $u \in V, v \in V'$:
$$\text{GED}(G, G') = \min_{(u,v) \in V \times V'} \text{GED}(G \backslash \{u\}, G' \backslash \{v\}) + c(u, v),$$

where $c(u, v)$ includes the cost of transforming $u \to v$ and modifying their incident edges.

For any node pair $(u, v)$ with degrees $d_u$ and $d_v$, the transformation can proceed in either of the two ways:

**(I) Node deletion and addition**: Delete $u$ and its incident edges (cost: $c_{ND} + d_u c_{ED}$), then add $v$ and its incident edges (cost: $c_{NA} + d_v c_{EA}$), yielding:
$$C_{\text{Add+Del}} = (c_{ND} + c_{NA}) + d_u c_{ED} + d_v c_{EA}.$$

**(II) Node substitution**: Substitute $u \to v$ (cost $c_{NS}$) and adjust edge differences:

$$C_{Sub} = c_{NS} + \max(d_u - d_v, 0)c_{ED} + \max(d_v - d_u, 0)c_{EA}.$$

Since $c_{NS} < c_{NA} + c_{ND}$, we have $C_{Sub} < C_{Add+Del}$. Thus, any edit path containing both a node addition and deletion can be replaced with a lower-cost substitution. This contradicts the optimality of such a path. Therefore, no optimal edit path includes both operations. $\square$

**Theorem 4.2.** *For any given pair of graphs $G = (V, E)$ and $G' = (V', E')$, the set of optimal edit paths $\mathcal{E}^*(G, G')$ is identical for all cost settings satisfying $c_{NS} = 0$ and $c_{NA}, c_{ND}, c_{EA}, c_{ED} > 0$*

*Proof.* Assume unpadded graphs and $|V| < |V'|$. By Proposition 4.1, under the condition that $c_{NS} = 0$, no optimal edit path includes both node additions and deletions. Consequently, to equalize the number of nodes from $G$ to $G'$, only node additions are required, with cost $C_{nodes} = c_{NA}(|V'| - |V|)$. Similarly, if $|V| > |V'|$, only node deletions occur, yielding a cost of $C_{nodes} = c_{ND}(|V| - |V'|)$.

To minimize the number of edge additions ($\#E_{add}$) and deletions ($\#E_{del}$), we solve the quadratic assignment problem:

$$\min_{\boldsymbol{P} \in \Pi_N} c_{ED} \cdot \#E_{del} + c_{EA} \cdot \#E_{add},$$

where: $\#E_{del} = \|[\boldsymbol{A} - \boldsymbol{P}\boldsymbol{A}'\boldsymbol{P}^\top]_+\|_{1,1}$ and $\#E_{add} = \|[\boldsymbol{P}\boldsymbol{A}'\boldsymbol{P}^\top - \boldsymbol{A}]_+\|_{1,1}$.

Since $c_{EA}, c_{ED} > 0$, minimizing $\#E_{add} + \#E_{del}$ ensures the total cost is minimized, regardless of the specific values of $c_{EA}$ and $c_{ED}$. The resulting $\#E_{add}^*$ and $\#E_{del}^*$ depend only on the structure of $G$ and $G'$, not the relative costs. Thus, the set of optimal edit paths $\mathcal{E}^*(G, G')$ remains the same across all valid cost settings satisfying $c_{NS} < c_{NA} + c_{ND}$ and $c_{NA}, c_{ND}, c_{NS}, c_{EA}, c_{ED} > 0$. $\square$

### 4.2. Implications

Theorem 4.2 establishes that the set of optimal edit paths $\mathcal{E}^*(G, G')$, or equivalently the optimal node alignment maps $\boldsymbol{P}^* \in \Pi_N$, depends only on the graph structures $G$ and $G'$ and remains invariant across a broad range of edit cost settings. While the GED *value* varies with the costs $c_{NA}, c_{ND}, c_{NS}, c_{EA}, c_{ED}$, the corresponding optimal *edit paths* remain unchanged. This enables us to (1) generate datasets of graph pairs $(G, G')$ with ground truth GED values efficiently computed across varying cost settings, and (2) construct adversarial tests to evaluate whether neural models correctly learn $\boldsymbol{P}^*$, ensuring alignment with interpretable edit paths.

## 5. Dataset Challenges and Train-Test Leakage

In this section, we examine the evolution of GED datasets (Section 5.1), the prevalence of structural redundancies (Section 5.2), and the impact of resulting train-test leakage on model evaluation (Section 5.3). Sections 5.4 and 5.5 build on the invariance result from Section 4, realizing its two key implications—scalable augmentation and adversarial testing—as targeted responses. All code and datasets used in this work have been made publicly available at https://anonymous.4open.science/r/better-graph-matching-7146/.

### 5.1. Evolution of GED Datasets and Their Limitations

The high computational expense of exact GED computation has significantly shaped the design of datasets used in neural GED research. Early models, such as SimGNN (Bai et al., 2019a), relied on small graph datasets like AIDS, LINUX, and IMDB, where most graphs contained at most 10 nodes. Exact GED values for these datasets were computed using the A* (Hart et al., 1968; Riesen et al., 2013) algorithm, which, while effective for small graphs, is not scalable to larger graphs. Subsequent models, including GraphSim (Bai et al., 2020), GEDGNN (Piao et al., 2023), EGSC (Qin et al., 2021), GENN-A* (Wang et al., 2021), GREED (Ranjan et al., 2022), and ERIC (Zhuo & Tan, 2022), continued to rely on these datasets. These datasets were later made available via PyTorch Geometric's GEDDataset class[1]. Although some of the later works extended to additional datasets, such as ALKANE (Bougleux et al., 2015), WILLOW (Cho et al., 2013), and NCI109 (Bai et al., 2019b), these too were largely composed of small-scale graphs with at most 10 nodes. As a result, the progress of neural GED research has been constrained by the limited size and scope of these datasets.

To address this, some papers, such as GMN (Li et al., 2019), explored synthetic datasets with graphs containing up to 50 nodes. These datasets were generated by applying controlled perturbations, such as edge additions and deletions, to source graphs, providing noisy estimates of graph distances. While this approach demonstrated scalability, it fell short of offering diverse, real-world graph structures. Similarly, the ICPR 2016 Graph Distance Contest[2] introduced larger graphs by using approximate GED solvers, such as BEAM (Neuhaus et al., 2006), HUNGARIAN (Riesen & Bunke, 2009b), and VJ (Fankhauser et al., 2011). The minimum cost from these solvers was used as an upper bound for exact GED. Although this enabled approximate GED computation for larger graphs, it sacrificed the precision needed for rigorous training and evaluation of neural GED

---

[1]https://pytorch-geometric.readthedocs.io/en/2.5.3/generated/torch_geometric.datasets.GEDDataset.html

[2]https://gdc2016.greyc.fr/

*Table 1.* Statistics of popular GED datasets and leakage analysis in standard train-test splits. Shown are the % Reduction ($\downarrow$ %) in the number of graphs after removing duplicates, as well as leakage rates for Intra-test-pairs (test pairs within $\mathcal{D}_{\text{test}}$) and Cross-train-test-pairs (test graphs paired with training graphs).

| | LINUX | IMDB | AIDS (w label) | AIDS (w/o label) |
|---|---|---|---|---|
| **Overall Dataset** | | | | |
| Number of graphs ($|\mathcal{D}|$) | 1000 | 1500 | 700 | 700 |
| Unique graphs ($|\mathcal{D}_{\text{unq}}|$) | 89 | 619 | 673 | 449 |
| % Reduction ($\downarrow$ %) | 91.1 | 58.7 | 3.8 | 35.8 |
| **Intra-test-pairs** | | | | |
| # Test pairs | 20100 | 45150 | 9870 | 9870 |
| # Leaked pairs | 18350 | 26463 | 24 | 4268 |
| % Leakage | 91.3 | 34.1 | 0.24 | 21.1 |
| **Cross-train-test-pairs** | | | | |
| # Test pairs | 120000 | 270000 | 58800 | 58800 |
| # Leaked pairs | 113716 | 148107 | 1721 | 23958 |
| % Leakage | 94.8 | 54.8 | 2.92 | 40.7 |

models.

A further limitation of these datasets stems from the uniform cost setting used for GED computation, where $c_{\text{NA}} = c_{\text{ND}} = c_{\text{EA}} = c_{\text{ED}} = 1$. This uniformity undermines the flexibility of the GED framework, which is capable of modeling both symmetric and asymmetric distances. Indeed, the ability of GED to stand as surrogate for other useful notions, such as subgraph isomorphism, is based on exploiting its general nature. Lastly, many studies do not release their generated datasets, resulting in repeated reliance on the original datasets introduced by SimGNN.

### 5.2. Assessing Train-Test Leakage

The three widely used GED datasets—LINUX, IMDB, and AIDS—consist of undirected, connected graphs, most with up to 10 nodes. LINUX and IMDB are unlabelled, while AIDS includes node labels. These datasets are divided into standardized training ($\mathcal{D}_{\text{train}}$) and test ($\mathcal{D}_{\text{test}}$) sets. Training supervision is achieved by pairing graphs in $\mathcal{D}_{\text{train}}$ and computing ground truth GED values using combinatorial solvers. For testing, two common schemes employed in prior works are: **(1)** *Intra-test-pairs*, which pairs all combinations of graphs within $\mathcal{D}_{\text{test}}$ ($O(|\mathcal{D}_{\text{test}}|^2)$ pairs) to evaluate generalization to unseen graph pairs from the same distribution; and **(2)** *Cross-train-test-pairs*, which pairs each graph in $\mathcal{D}_{\text{test}}$ with all graphs in $\mathcal{D}_{\text{train}}$ to assess GED computation for new queries against a fixed corpus, which is particularly useful for retrieval setups.

Our analysis of these datasets reveals a significant number of structurally isomorphic graphs. Modern GNN-based neural GED models, being permutation-invariant, produce identical embeddings for graphs within the same isomorphism class, provided they share identical initializations.. Con-

sequently, the number of unique graphs ($\mathcal{D}_{\text{unq}}$) is reduced to a small fraction of the original dataset size, as shown in Table 1. This structural redundancy results in substantial train-test leakage, manifesting differently across testing schemes: in Intra-test-pairs, isomorphic graphs split between $\mathcal{D}_{\text{train}}$ and $\mathcal{D}_{\text{test}}$ lead to graph pairs within $\mathcal{D}_{\text{test}}$ which are already seen during training, causing leakage as high as 91.3%, as shown in Table 1; in Cross-train-test-pairs, test graphs already encountered during training are paired with training graphs, resulting in memorized pairs and leakage rates as high as 94.8%, as shown in Table 1.

### 5.3. Implications

In Table 2, we quantify the impact of train-test leakage on baseline models using Intra-test-pairs and Cross-train-test-pairs. We evaluate these models on the default dataset test splits, which include leakage, and report the Mean Squared Error (MSE) and Kendall Tau Correlation (Ktau) between predicted and ground truth GED values under both Intra-test-pairs and Cross-train-test-pairs. Furthermore, to address the issue of train-test leakage, we apply a rigorous graph pair generation strategy (Algorithm 3, GENERATEPAIRS). Given the benchmark dataset $\mathcal{D}$, we perform an explicit isomorphism check for all graph pairs, filtering out isomorphic graphs to obtain a set of unique graphs $\mathcal{S}$. This unique set is split into $\mathcal{S}_{\text{train}}, \mathcal{S}_{\text{val}}$, and $\mathcal{S}_{\text{test}}$ in a 60:20:20 ratio. For each split, graph pairs $\mathcal{P}_{\bullet}$ are generated by considering all combinations (including self-pairs), yielding $|\mathcal{S}_{\bullet}| \times (|\mathcal{S}_{\bullet}| + 1)/2$ pairs for Intra-test-pairs. For Cross-train-test-pairs, all test graphs are paired with all training graphs. The ground truth GED values $\mathcal{Y}_{\bullet}$ for these pairs are computed using the MIP-F2 solver (Lerouge et al., 2017). By ensuring that all graphs within and across splits are unique, this pipeline effectively eliminates isomorphic leakage. A schematic of this process is shown in Figure 1.

**Observations:** **(1)** Models perform significantly better *with leakage*, particularly on LINUX, benefiting from memorization rather than generalization. Removing leakage increases MSE and lowers Ktau across all datasets, exposing inflated performance. **(2)** ISONET, GraphSim, and GEDGNN show the largest decline; ISONET's MSE on LINUX rises from 0.323 to 5.146, with Ktau dropping from 0.903 to 0.646. Originally designed for subgraph matching, ISONET benefits disproportionately from leakage. **(3)** The removal of redundancy makes generalization differences more apparent, as seen in the widened Ktau gaps for LINUX. **(4)** Lower redundancy in this dataset results in minimal impact from leakage removal, suggesting less overfitting to memorized graph structures. **(5)** While there are no major ranking reversals among models, the considerable drop in performance across all architectures indicates that existing neural models still struggle with generalization. We present further discussion based on Cross-train-test-pairs results in Appendix A.

*Table 2.* Mean Squared Error (MSE) and Kendall's Tau (Ktau) correlation for GED prediction models on standard datasets before and after train-test leakage removal. We report results across Intra-test-pairs and Cross-train-test-pairs. For each setting, models are evaluated under two conditions: (i) **With Leakage**—where dataset leakage exists due to isomorphic graphs in both train and test splits, and (ii) **Without Leakage**—where leakage is removed via dataset preprocessing.

| | MSE (Lower is Better) | | | | | | Ktau (Higher is Better) | | | | | |
| --- | --- | --- | --- | --- | --- | --- | --- | --- | --- | --- | --- | --- |
| | With Leakage | | | Without Leakage | | | With Leakage | | | Without Leakage | | |
| | LINUX | AIDS (w label) | AIDS (w/o label) | LINUX | AIDS (w label) | AIDS (w/o label) | LINUX | AIDS (w label) | AIDS (w/o label) | LINUX | AIDS (w label) | AIDS (w/o label) |
| **Intra-test-pairs** | | | | | | | | | | | | |
| GMN-Match | 0.062 | 0.811 | 0.558 | 0.823 | 0.825 | 0.635 | 0.937 | 0.785 | 0.798 | 0.793 | 0.775 | 0.736 |
| GMN-Embed | 0.219 | 1.525 | 0.578 | 1.238 | 1.575 | 0.669 | 0.917 | 0.718 | 0.800 | 0.745 | 0.701 | 0.729 |
| SimGNN | 0.277 | 1.180 | 0.879 | 2.350 | 1.406 | 0.899 | 0.903 | 0.744 | 0.752 | 0.642 | 0.720 | 0.688 |
| GraphSim | 0.165 | 1.494 | 1.115 | 2.511 | 1.345 | 1.371 | 0.921 | 0.700 | 0.703 | 0.599 | 0.708 | 0.599 |
| GREED | 0.203 | 1.409 | 0.614 | 1.415 | 1.748 | 0.754 | 0.921 | 0.721 | 0.788 | 0.755 | 0.677 | 0.710 |
| GEDGNN | 7.463 | 9.069 | 7.832 | 6.018 | 8.551 | 4.778 | 0.296 | 0.039 | 0.022 | 0.066 | 0.046 | -0.052 |
| ISONET | 0.323 | 0.995 | 0.837 | 5.146 | 1.032 | 0.978 | 0.903 | 0.765 | 0.746 | 0.646 | 0.749 | 0.669 |
| H2MN | 0.244 | 1.350 | 0.785 | 2.470 | 1.487 | 1.021 | 0.909 | 0.729 | 0.761 | 0.571 | 0.703 | 0.673 |
| EGSC | 0.071 | 0.993 | 0.438 | 3.297 | 1.135 | 0.562 | 0.937 | 0.761 | 0.829 | 0.707 | 0.743 | 0.761 |
| ERIC | 0.049 | 1.255 | 0.437 | 1.013 | 1.369 | 0.533 | 0.940 | 0.739 | 0.832 | 0.796 | 0.718 | 0.767 |
| GRAPHEDX | 0.052 | 1.647 | 0.801 | 0.827 | 1.931 | 0.872 | 0.940 | 0.694 | 0.771 | 0.844 | 0.682 | 0.714 |
| **Cross-train-test-pairs** | | | | | | | | | | | | |
| GMN-Match | 0.047 | 0.743 | 0.510 | 0.689 | 0.757 | 0.582 | 0.934 | 0.806 | 0.815 | 0.802 | 0.799 | 0.767 |
| GMN-Embed | 0.202 | 1.265 | 0.531 | 0.925 | 1.225 | 0.599 | 0.913 | 0.752 | 0.814 | 0.766 | 0.748 | 0.766 |
| SimGNN | 0.282 | 1.051 | 0.851 | 1.593 | 1.127 | 0.870 | 0.892 | 0.765 | 0.758 | 0.629 | 0.746 | 0.708 |
| GraphSim | 0.121 | 1.296 | 1.055 | 1.618 | 1.218 | 1.264 | 0.921 | 0.730 | 0.721 | 0.626 | 0.739 | 0.630 |
| GREED | 0.197 | 1.169 | 0.571 | 1.326 | 1.442 | 0.696 | 0.916 | 0.758 | 0.803 | 0.703 | 0.725 | 0.743 |
| GEDGNN | 6.721 | 7.453 | 7.337 | 5.247 | 7.595 | 5.663 | 0.324 | 0.091 | 0.051 | 0.095 | 0.055 | -0.048 |
| ISONET | 0.286 | 0.905 | 0.707 | 4.427 | 0.937 | 0.854 | 0.900 | 0.781 | 0.775 | 0.656 | 0.770 | 0.708 |
| H2MN | 0.247 | 1.159 | 0.755 | 1.712 | 1.405 | 0.928 | 0.900 | 0.755 | 0.769 | 0.593 | 0.718 | 0.692 |
| EGSC | 0.059 | 0.865 | 0.375 | 3.305 | 0.934 | 0.516 | 0.933 | 0.790 | 0.847 | 0.713 | 0.778 | 0.788 |
| ERIC | 0.040 | 1.007 | 0.381 | 0.939 | 1.065 | 0.468 | 0.935 | 0.777 | 0.848 | 0.758 | 0.765 | 0.800 |
| GRAPHEDX | 0.046 | 1.293 | 0.700 | 0.624 | 1.498 | 0.769 | 0.935 | 0.739 | 0.792 | 0.833 | 0.726 | 0.749 |

## 5.4. Efficient Training Data Augmentation

Given two graphs $G$ and $G'$, suppose a combinatorial solver provides an optimal node alignment matrix $P \in \Pi_N$. As shown in Theorem 4.2, $P$ remains the same across all cost settings $\mathcal{C} = \{c_{NS}, c_{NA}, c_{ND}, c_{EA}, c_{ED}\}$ satisfying $c_{NS} = 0$ and $c_{NS}, c_{NA}, c_{ND}, c_{EA}, c_{ED} > 0$, with

$$\text{GED}_{\mathcal{C}}(G, G') = c_{ND} \cdot \#N_{del} + c_{NA} \cdot \#N_{add} +$$
$$c_{NS} \cdot \#N_{sub} + c_{ED} \cdot \#E_{del} + c_{EA} \cdot \#E_{add}, \quad \text{where}$$

$$\#N_{del} = \|[\mathbf{I}_G - \boldsymbol{P}\mathbf{I}_{G'}]_+\|_1,$$
$$\#N_{add} = \|[\boldsymbol{P}\mathbf{I}_{G'} - \mathbf{I}_G]_+\|_1,$$
$$\#E_{del} = \tfrac{1}{2}\|[\boldsymbol{A} - \boldsymbol{P}\boldsymbol{A}'\boldsymbol{P}^\top]_+\|_{1,1},$$
$$\#E_{add} = \tfrac{1}{2}\|[\boldsymbol{P}\boldsymbol{A}'\boldsymbol{P}^\top - \boldsymbol{A}]_+\|_{1,1}, \quad \text{and} \qquad (1)$$

$$\#N_{sub} = \sum_{i=1}^{|V|} \sum_{j=1}^{|V'|} \mathbf{I}_G[i] \cdot (\boldsymbol{P}\mathbf{I}_{G'})[j] \cdot \mathbb{I}[\ell_V(i) \neq \ell_V(j)].$$

We exploit this, proposing a structured pipeline (Algorithm 2) that eliminates leakage, optimizes computation, and generates multiple cost-specific datasets efficiently. First, Algorithm 3 (GENERATEPAIRS) removes isomorphic duplicates and partitions the dataset into disjoint splits, constructing leakage-free graph pairs. Next, Algorithm 4 (COMPUTEOPTIMALPATHS) computes the optimal node alignment $\boldsymbol{P}^*$ for all graph pairs using a fixed cost setting $\mathcal{C}_0$.

**Algorithm 2** Dataset Processing with Cost Variants

**Require:** $\mathcal{D}$ {Input dataset}
**Ensure:** $\{\Delta_{k,\text{train}}, \Delta_{k,\text{val}}, \Delta_{k,\text{Intra-test}}, \Delta_{k,\text{Cross-train-test}}\}_{k=1}^{K}$
1: $\mathcal{P}_{\text{train}}, \mathcal{P}_{\text{val}}, \mathcal{P}_{\text{Intra-test}}, \mathcal{P}_{\text{Cross-train-test}} \leftarrow$ GENERATEPAIRS$(\mathcal{D})$
2: **for each** $\mathcal{P}_\bullet \in \{\mathcal{P}_{\text{train}}, \mathcal{P}_{\text{val}}, \mathcal{P}_{\text{Intra-test}}, \mathcal{P}_{\text{Cross-train-test}}\}$ **do**
3: $\quad \mathcal{Q}_\bullet \leftarrow$ COMPUTEOPTIMALPATHS$(\mathcal{P}_\bullet)$
4: Let $\{\mathcal{C}_k\}_{k=1}^{K}$ be the set of compatible costs (Thm. 4.2)
5: **for each** $k \in [K]$ **do**
6: $\quad$ **for each** $\mathcal{Q}_\bullet \in \{\mathcal{Q}_{\text{train}}, \mathcal{Q}_{\text{val}}, \mathcal{Q}_{\text{Intra-test}}, \mathcal{Q}_{\text{Cross-train-test}}\}$ **do**
7: $\qquad \Delta_{k,\bullet} \leftarrow$ GENERATECOSTVARIANTS$(\mathcal{Q}_\bullet, \mathcal{C}_k)$
8: **return** $\{\Delta_{k,\text{train}}, \Delta_{k,\text{val}}, \Delta_{k,\text{Intra-test}}, \Delta_{k,\text{Cross-train-test}}\}_{k=1}^{K}$

Finally, Algorithm 5 (GENERATECOSTVARIANTS) reuses $\boldsymbol{P}^*$ to compute GED via Eq. (1) across varying compatible cost settings $\{\mathcal{C}_k\}_{k=1}^{K}$, producing multiple datasets without redundant alignment computations, ensuring scalable dataset generation while maintaining train-test separation.

## 5.5. Evaluation via Cost-Invariant Edit Paths

Existing neural GED models largely overlook variable cost settings. The only exception, GRAPHEDX, explores only two cases: equal cost

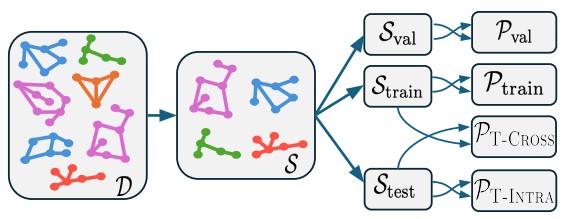 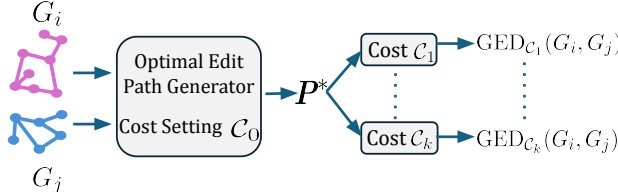

*Figure 1.* **Left:** The raw dataset $\mathcal{D}$ undergoes isomorphism filtering (Algorithm 3) to yield the unique set $\mathcal{S}$, which is then split into training ($\mathcal{S}_{\text{train}}$), validation ($\mathcal{S}_{\text{val}}$), and test ($\mathcal{S}_{\text{test}}$) sets. Graph pairs are constructed within training and validation splits, while test pairs follow two schemes: $\mathcal{P}_{\text{T-INTRA}}$ (Intra-test-pairs) and $\mathcal{P}_{\text{T-CROSS}}$ (Cross-train-test-pairs). **Right:** For each graph pair $(G_i, G_j)$, Algorithm 4 applies a combinatorial solver under a fixed cost setting $\mathcal{C}_0$ to compute the optimal node alignment $\boldsymbol{P}^*$. Algorithm 5 then leverages Theorem 4.2 to efficiently compute GED across multiple cost settings $\{\mathcal{C}_k\}_{k=1}^K$, exploiting the invariance of $\boldsymbol{P}^*$.

*Table 3.* Performance comparison in terms of MSE and Ktau evaluated on Code2 dataset for the models trained with $\mathcal{C}_0$ cost setting. Evaluation is done for different test cost configurations $\mathcal{C}_0$–$\mathcal{C}_4$ for Intra-test-pairs test scheme.

| | MSE (Lower is Better) | | | | | Ktau (Higher is Better) | | | | |
|---|---|---|---|---|---|---|---|---|---|---|
| | $\mathcal{C}_0$ | $\mathcal{C}_1$ | $\mathcal{C}_2$ | $\mathcal{C}_3$ | $\mathcal{C}_4$ | $\mathcal{C}_0$ | $\mathcal{C}_1$ | $\mathcal{C}_2$ | $\mathcal{C}_3$ | $\mathcal{C}_4$ |
| GMN-Match | 1.68 | 79.86 | 1002.01 | 10102.82 | 28743.35 | 0.88 | 0.75 | 0.52 | 0.77 | 0.84 |
| GMN-Embed | 1.36 | 80.79 | 1003.41 | 10128.47 | 28782.49 | 0.89 | 0.77 | 0.53 | 0.78 | 0.86 |
| SimGNN | 2.67 | 74.70 | 973.74 | 10063.38 | 28646.82 | 0.86 | 0.75 | 0.55 | 0.72 | 0.84 |
| GraphSim | 3.14 | 77.85 | 991.73 | 10043.54 | 28638.96 | 0.84 | 0.71 | 0.51 | 0.74 | 0.81 |
| GREED | 1.87 | 77.52 | 991.55 | 10092.37 | 28715.97 | 0.88 | 0.77 | 0.53 | 0.77 | 0.84 |
| GEDGNN | 31.12 | 161.77 | 1181.59 | 10722.40 | 29789.05 | -0.03 | -0.13 | -0.25 | 0.07 | -0.06 |
| ISONET | 0.88 | 75.49 | 999.90 | 10103.25 | 28738.95 | 0.92 | 0.79 | 0.54 | 0.79 | 0.89 |
| H2MN | 6.20 | 83.92 | 983.70 | 10123.29 | 28773.79 | 0.75 | 0.67 | 0.52 | 0.60 | 0.73 |
| EGSC | 4.16 | 94.54 | 1058.17 | 10190.29 | 28937.83 | 0.81 | 0.70 | 0.46 | 0.76 | 0.77 |
| ERIC | 1.36 | 84.53 | 1015.90 | 10156.43 | 28836.03 | 0.89 | 0.75 | 0.53 | 0.78 | 0.86 |
| GRAPHEDX | 0.76 | 76.90 | 2434.89 | 2102.35 | 688.71 | 0.92 | 0.78 | 0.52 | 0.74 | 0.89 |

---

**Algorithm 3** GENERATEPAIRS

**Require:** $\mathcal{D}$ {Input dataset}
**Ensure:** $\mathcal{P}_{\text{train}}, \mathcal{P}_{\text{val}}, \mathcal{P}_{\text{Intra-test-pairs}}, \mathcal{P}_{\text{Cross-train-test-pairs}}$
1: $\mathcal{S} \leftarrow$ Remove Isomorphic Duplicates from $\mathcal{D}$
2: $\mathcal{S}_{\text{train}}, \mathcal{S}_{\text{val}}, \mathcal{S}_{\text{test}} \leftarrow \text{Split}(\mathcal{S})$
3: $\mathcal{P}_{\text{train}} \leftarrow \{(G_i, G_j) \mid G_i, G_j \in \mathcal{S}_{\text{train}}\}$
4: $\mathcal{P}_{\text{val}} \leftarrow \{(G_i, G_j) \mid G_i, G_j \in \mathcal{S}_{\text{val}}\}$
5: $\mathcal{P}_{\text{Intra-test-pairs}} \leftarrow \{(G_i, G_j) \mid G_i, G_j \in \mathcal{S}_{\text{test}}\}$
6: $\mathcal{P}_{\text{Cross-train-test-pairs}} \leftarrow \{(G_i, G_j) \mid G_i \in \mathcal{S}_{\text{test}}, G_j \in \mathcal{S}_{\text{train}}\}$
7: **return** $\mathcal{P}_{\text{train}}, \mathcal{P}_{\text{val}}, \mathcal{P}_{\text{Intra-test-pairs}}, \mathcal{P}_{\text{Cross-train-test-pairs}}$

---

**Algorithm 4** COMPUTEOPTIMALPATHS

**Require:** $\mathcal{P}$ {Set of graph pairs $(G_i, G_j)$}
**Ensure:** $\mathcal{Q} = \{(G_i, G_j, \boldsymbol{P}^*) \mid (G_i, G_j) \in \mathcal{P}_\bullet\}$
1: $\mathcal{Q} \leftarrow \emptyset$
2: $\mathcal{C}_0 = \{c_{\text{NS}} = 0, c_{\text{NA}} = 1, c_{\text{ND}} = 1, c_{\text{EA}} = 1, c_{\text{ED}} = 1\}$
3: **for each** $(G_i, G_j) \in \mathcal{P}$ **do**
4: $\quad \boldsymbol{P}^* \leftarrow$ Obtain optimal node alignment with $\mathcal{C}$
5: $\quad \mathcal{Q} \leftarrow \mathcal{Q} \cup \{(G_i, G_j, \boldsymbol{P}^*)\}$
6: **return** $\mathcal{Q}$

---

**Algorithm 5** GENERATECOSTVARIANTS

**Require:** $\mathcal{Q}$ {Set of graph pairs with $\boldsymbol{P}^*$}
**Ensure:** $\{\Delta_k\}_{k=1}^K$ – multiple datasets under varying $\mathcal{C}_k$
1: **for each** $k \in [K]$ **do**
2: $\quad \Delta_k \leftarrow \emptyset$
3: $\quad$ **for each** $(G_i, G_j, \boldsymbol{P}^*) \in \mathcal{Q}$ **do**
4: $\quad\quad$ Compute $\text{GED}_{\mathcal{C}_k}(G_i, G_j)$ using Eq. (1)
5: $\quad\quad \Delta_k \leftarrow \Delta_k \cup \{(G_i, G_j, \text{GED}_{\mathcal{C}_k})\}$
6: **return** $\{\Delta_k\}_{k=1}^K$

---

($\mathcal{C}_0$: $\{c_{\text{NA}}, c_{\text{ND}}, c_{\text{NS}}, c_{\text{EA}}, c_{\text{ED}}\} = (1, 1, 0, 1, 1)$) and unequal cost ($\mathcal{C}_1$: $\{c_{\text{NA}}, c_{\text{ND}}, c_{\text{NS}}, c_{\text{EA}}, c_{\text{ED}}\} = (1, 3, 0, 1, 2)$), training separate models for each without generalizing further. Moreover, whether its predicted alignments approximate a ground-truth edit path remains unclear.

Building on Theorem 4.2, we create an **adversarial test set** to assess whether models recover optimal node alignments ($\boldsymbol{P}^*$) or overfit to dataset-specific patterns. Using four leakage-free datasets (Mutag, Code2, Molhiv, Molpcba) from GRAPHEDX, we generate test sets across five additional cost settings ($\mathcal{C}_2$–$\mathcal{C}_6$) (Table 5, Appendix A). As

before, we evaluate MSE and Ktau under Intra-test-pairs ($\mathcal{C}_0$–$\mathcal{C}_4$) here, with full results in Appendix A.

We compare model performance under a fixed cost setting

using MSE, noting that MSE values across different cost settings are not directly comparable due to varying cost magnitudes. However, Ktau effectively captures the variance in model rankings across cost settings. We observe: **(1)** Most models exhibit significant Ktau variation, with drops of up to 0.4, highlighting their limited generalization to unseen cost settings. While GRAPHEDXperforms best on $\mathcal{C}_0$, other baselines achieve comparable performance across higher-cost settings. Ground truth rankings can shift significantly with cost variations, but models trained on fixed-cost settings fail to adapt, producing unchanged predictions that misalign with the new rank order. **(2)** Most models, which focus solely on node alignment, achieve lower MSE on $\mathcal{C}_2$ compared to $\mathcal{C}_3$ or $\mathcal{C}_4$, both of which involve high edge edit costs (Table 5). The exception is GRAPHEDX, which explicitly aligns both nodes and edges, resulting in significantly lower MSE on $\mathcal{C}_3$ and $\mathcal{C}_4$.

## 6. Alternative Views

In real-world applications, structurally similar graphs may naturally recur between training and inference phases. This raises the question of whether such repetition constitutes a flaw or simply reflects the nature of practical data distributions.

In practical applications, a graph matching system may be tasked with two distinct kinds of inference: *rote recall*, where the test-time input is structurally identical (or nearly so) to data seen during training, and *generalization*, where the system must handle unseen or structurally different graph instances. While rote recall can often be addressed through simple caching or lookup tables, generalization requires true learning. As such, it is critical that benchmark evaluations clearly distinguish between these two modes of inference.

However, due to the lack of fine-grained control over test fold construction in existing datasets, these cases are frequently conflated in reported performance. This has led to inflated results that obscure a model's actual generalization capabilities. We argue that this conflation undermines the primary goal of learning-based graph matching: robust generalization across structurally diverse inputs.

Our position is that train-test leakage—such as isomorphic or near-isomorphic graphs appearing across splits—is not merely a benign artifact of real-world repetition, but a fundamental flaw in benchmarking design when left unchecked. In practice, structural overlap between training and test data can artificially inflate evaluation metrics, masking poor generalization and discouraging model innovations.

This perspective is consistent with recent work in graph learning that emphasizes structural diversity during both training and evaluation (Mahdavi et al., 2022; Velikonivtsev

et al., 2024). These works highlight how graph-level generalization is sensitive to structure-induced biases in data generation. Our proposed benchmark framework—centered on variable-cost GED, alignment-sensitive evaluation, and isomorphism-free test construction—provides a principled path forward. By decoupling rote recall from genuine generalization, we aim to foster more faithful, reproducible, and actionable evaluation protocols for neural graph matching systems.

## 7. Conclusion

We rigorously analyze the limitations of existing neural GED benchmarks, identifying significant train-test leakage and proposing a robust dataset construction pipeline to mitigate it. Even if stopping leakage does not completely overthrow broad comparisons between methods, leakage prevention is critical to understand how learning takes place and what to expect in the face of domain shifts. Leveraging Theorem 4.2, we introduce a principled approach for generating large-scale training datasets and designing adversarial test sets to evaluate whether models recover meaningful edit paths. Our empirical analysis under varying cost settings reveals that current neural models struggle with generalization, emphasizing the need for more robust architectures. These insights pave the way for future research in designing cost-aware, interpretable, and generalizable GED models.

## Impact Statement

This work contributes to the advancement of GED estimation by addressing limitations in current benchmark datasets and evaluation protocols. Our findings highlight the challenges of generalization under varying cost settings and propose improved methodologies for assessing model robustness. While this research primarily focuses on methodological improvements within machine learning, it has broader implications for applications in bioinformatics, cheminformatics, and network analysis, where accurate GED estimation is crucial.

We do not foresee direct ethical concerns arising from this work. However, as GED-based methods are used in domains such as fraud detection and security analysis, it is important to ensure that models trained on biased or incomplete data do not lead to unintended consequences.

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

# A. Additional Experiments

We provide additional evaluation results beyond those presented in the main paper.

## A.1. Additional evaluation of impact of leakage as per Cross-train-test-pairs

In Table 2 of main paper, we presented results for Intra-test-pairs ans Cross-train-test-pairs. Here, we provide additional observations based on results for Cross-train-test-pairs, where each test graph is paired with all training graphs. This setup evaluates the generalization of models when comparing unseen test graphs against a fixed reference set. Table 4 reports the Mean Squared Error (MSE) and Kendall's Tau (Ktau) correlation for models under Cross-train-test-pairs, both before and after leakage removal.

*Table 4.* Mean Squared Error (MSE) and Kendall's Tau (Ktau) correlation for GED prediction models on standard datasets before and after train-test leakage removal. We report results across Intra-test-pairs (intra-test pairs) and Cross-train-test-pairs (cross-train-test pairs). For each setting, models are evaluated under two conditions: (i) **With Leakage**—where dataset leakage exists due to isomorphic graphs in both train and test splits, and (ii) **Without Leakage**—where leakage is removed via dataset preprocessing.

| | MSE (Lower is Better) | | | | | | Ktau (Higher is Better) | | | | | |
| | With Leakage | | | Without Leakage | | | With Leakage | | | Without Leakage | | |
| | LINUX | AIDS (w label) | AIDS (w/o label) | LINUX | AIDS (w label) | AIDS (w/o label) | LINUX | AIDS (w label) | AIDS (w/o label) | LINUX | AIDS (w label) | AIDS (w/o label) |
|---|---|---|---|---|---|---|---|---|---|---|---|---|
| | | | | | | **Intra-test-pairs** | | | | | | |
| GMN-Match | 0.062 | 0.811 | 0.558 | 0.823 | 0.825 | 0.635 | 0.937 | 0.785 | 0.798 | 0.793 | 0.775 | 0.736 |
| GMN-Embed | 0.219 | 1.525 | 0.578 | 1.238 | 1.575 | 0.669 | 0.917 | 0.718 | 0.800 | 0.745 | 0.701 | 0.729 |
| SimGNN | 0.277 | 1.180 | 0.879 | 2.350 | 1.406 | 0.899 | 0.903 | 0.744 | 0.752 | 0.642 | 0.720 | 0.688 |
| GraphSim | 0.165 | 1.494 | 1.115 | 2.511 | 1.345 | 1.371 | 0.921 | 0.700 | 0.703 | 0.599 | 0.708 | 0.599 |
| GREED | 0.203 | 1.409 | 0.614 | 1.415 | 1.748 | 0.754 | 0.921 | 0.721 | 0.788 | 0.755 | 0.677 | 0.710 |
| GEDGNN | 7.463 | 9.069 | 7.832 | 6.018 | 8.551 | 4.778 | 0.296 | 0.039 | 0.022 | 0.066 | 0.046 | -0.052 |
| ISONET | 0.323 | 0.995 | 0.837 | 5.146 | 1.032 | 0.978 | 0.903 | 0.765 | 0.746 | 0.646 | 0.749 | 0.669 |
| H2MN | 0.244 | 1.350 | 0.785 | 2.470 | 1.487 | 1.021 | 0.909 | 0.729 | 0.761 | 0.571 | 0.703 | 0.673 |
| EGSC | 0.071 | 0.993 | 0.438 | 3.297 | 1.135 | 0.562 | 0.937 | 0.761 | 0.829 | 0.707 | 0.743 | 0.761 |
| ERIC | 0.049 | 1.255 | 0.437 | 1.013 | 1.369 | 0.533 | 0.940 | 0.739 | 0.832 | 0.796 | 0.718 | 0.767 |
| GRAPHEDX | 0.052 | 1.647 | 0.801 | 0.827 | 1.931 | 0.872 | 0.940 | 0.694 | 0.771 | 0.844 | 0.682 | 0.714 |
| | | | | | | **Cross-train-test-pairs** | | | | | | |
| GMN-Match | 0.047 | 0.743 | 0.510 | 0.689 | 0.757 | 0.582 | 0.934 | 0.806 | 0.815 | 0.802 | 0.799 | 0.767 |
| GMN-Embed | 0.202 | 1.265 | 0.531 | 0.925 | 1.225 | 0.599 | 0.913 | 0.752 | 0.814 | 0.766 | 0.748 | 0.766 |
| SimGNN | 0.282 | 1.051 | 0.851 | 1.593 | 1.127 | 0.870 | 0.892 | 0.765 | 0.758 | 0.629 | 0.746 | 0.708 |
| GraphSim | 0.121 | 1.296 | 1.055 | 1.618 | 1.218 | 1.264 | 0.921 | 0.730 | 0.721 | 0.626 | 0.739 | 0.630 |
| GREED | 0.197 | 1.169 | 0.571 | 1.326 | 1.442 | 0.696 | 0.916 | 0.758 | 0.803 | 0.703 | 0.725 | 0.743 |
| GEDGNN | 6.721 | 7.453 | 7.337 | 5.247 | 7.595 | 5.663 | 0.324 | 0.091 | 0.051 | 0.095 | 0.055 | -0.048 |
| ISONET | 0.286 | 0.905 | 0.707 | 4.427 | 0.937 | 0.854 | 0.900 | 0.781 | 0.775 | 0.656 | 0.770 | 0.708 |
| H2MN | 0.247 | 1.159 | 0.755 | 1.712 | 1.405 | 0.928 | 0.900 | 0.755 | 0.769 | 0.593 | 0.718 | 0.692 |
| EGSC | 0.059 | 0.865 | 0.375 | 3.305 | 0.934 | 0.516 | 0.933 | 0.790 | 0.847 | 0.713 | 0.778 | 0.788 |
| ERIC | 0.040 | 1.007 | 0.381 | 0.939 | 1.065 | 0.468 | 0.935 | 0.777 | 0.848 | 0.758 | 0.765 | 0.800 |
| GRAPHEDX | 0.046 | 1.293 | 0.700 | 0.624 | 1.498 | 0.769 | 0.935 | 0.739 | 0.792 | 0.833 | 0.726 | 0.749 |

The trends observed in Intra-test-pairs extend to Cross-train-test-pairs, reinforcing key insights:

(1) **Effect of Train-Test Leakage:** Models perform significantly better *with leakage*, particularly on LINUX, which has the highest redundancy. This indicates that models rely on memorization rather than generalization. After leakage removal, MSE increases and Ktau declines across all datasets, demonstrating that previously inflated performance was largely an artifact of structural duplication in the test set.

(2) **Performance Drop Post-Leakage Removal:** The decline is most severe for ISONET, GraphSim, and GEDGNN. For instance in Cross-train-test-pairs, ISONET's MSE on LINUX rises from 0.286 to 4.427, while Ktau falls from 0.9 to 0.656. This is expected, as ISONETwas originally designed for subgraph matching. Leakage previously enabled overfitting, creating an illusion of strong performance.

(3) **Increased Disparity Across Models:** Without leakage, performance differences between models become more pronounced as well in Cross-train-test-pairs, particularly in Ktau scores for LINUX. This provides further evidence that, once deprived of memorization advantages, models reveal their true generalization capabilities, leading to a clearer performance ranking.

(4) **Lower Impact on AIDS (w/ labels):** We observe once again in Cross-train-test-pairs that the effect of leakage removal

is milder for AIDS (w/ labels) due to lower structural redundancy. Models trained on datasets with fewer isomorphic graphs are naturally less susceptible to memorization-driven overfitting.

(5) **Consistent Model Ranking but Substantial Performance Drop:** While model rankings remain largely unchanged, the overall performance decline across both Intra-test-pairs and Cross-train-test-pairs highlights the fundamental difficulty of learning GED. These findings emphasize the need for more rigorous evaluation benchmarks and improved model architectures.

### A.2. Adversarial Evaluation via Cost-Invariant Edit Paths under Varying Cost Settings

#### A.2.1. COST CONFIGURATIONS FOR GED EVALUATION

Table 5 lists the different cost settings $\mathcal{C}_k$ used for GED computation. The first two, $\mathcal{C}_0$ (equal cost) and $\mathcal{C}_1$ (unequal cost), were introduced in GRAPHEDX. These settings assume a fixed penalty for node and edge modifications but do not explore broader variations in cost configurations.

Leveraging Theorem 4.2, we extend this analysis by introducing additional cost settings ($\mathcal{C}_2$–$\mathcal{C}_6$), covering a diverse range of node and edge operation costs. Our theorem guarantees that, as long as $c_{\text{NS}} < c_{\text{NA}} + c_{\text{ND}}$, the optimal edit path (and corresponding node alignment $\boldsymbol{P}^*$) remains invariant across these cost settings. This property allows us to generate adversarial test cases that assess whether neural models recover meaningful edit paths rather than simply memorizing dataset-specific patterns.

By evaluating models under varying cost configurations, we test their robustness in predicting GED and their ability to generalize beyond the specific cost assumptions made during training.

*Table 5.* Cost settings $\mathcal{C}_k$ used for GED computation.

| $\mathcal{C}_k$ | $c_{\text{NA}}$ | $c_{\text{ND}}$ | $c_{\text{NS}}$ | $c_{\text{EA}}$ | $c_{\text{ED}}$ |
|---|---|---|---|---|---|
| $\mathcal{C}_0$ | 1 | 1 | 0 | 1 | 1 |
| $\mathcal{C}_1$ | 1 | 3 | 0 | 1 | 2 |
| $\mathcal{C}_2$ | 10 | 15 | 0 | 1 | 1 |
| $\mathcal{C}_3$ | 1 | 1 | 0 | 11 | 12 |
| $\mathcal{C}_4$ | 18 | 23 | 0 | 19 | 12 |
| $\mathcal{C}_5$ | 21 | 14 | 0 | 20 | 26 |
| $\mathcal{C}_6$ | 2 | 29 | 0 | 26 | 3 |

A.2.2. EXTENDED OBSERVATIONS ON VARYING COST TEST SETS

In Table 3, we analyzed model performance under a fixed cost setting using MSE and Ktau, highlighting that MSE values are not directly comparable across cost settings due to varying cost magnitudes. However, Ktau effectively captures shifts in model rankings across different test configurations. Here in Table 6, we extend these observations by incorporating results across all seven cost settings ($\mathcal{C}_0$–$\mathcal{C}_6$) and including evaluations under Cross-train-test-pairs.

*Table 6.* Performance comparison in terms of MSE and Ktau evaluated on **Code2** dataset for the models trained with $\mathcal{C}_0$ cost setting. Evaluation is done for different test cost configurations $\mathcal{C}_0$–$\mathcal{C}_6$ across both Intra-test-pairs and Cross-train-test-pairs test schemes.

| | MSE (Lower is Better) | | | | | | | Ktau (Higher is Better) | | | | | | |
|---|---|---|---|---|---|---|---|---|---|---|---|---|---|---|
| | $\mathcal{C}_0$ | $\mathcal{C}_1$ | $\mathcal{C}_2$ | $\mathcal{C}_3$ | $\mathcal{C}_4$ | $\mathcal{C}_5$ | $\mathcal{C}_6$ | $\mathcal{C}_0$ | $\mathcal{C}_1$ | $\mathcal{C}_2$ | $\mathcal{C}_3$ | $\mathcal{C}_4$ | $\mathcal{C}_5$ | $\mathcal{C}_6$ |
| **Intra-test-pairs** | | | | | | | | | | | | | | |
| GMN-Match | 1.68 | 79.86 | 1002.01 | 10102.82 | 28743.35 | 55339.54 | 23766.56 | 0.88 | 0.75 | 0.52 | 0.77 | 0.84 | 0.84 | 0.84 |
| GMN-Embed | 1.36 | 80.79 | 1003.41 | 10128.47 | 28782.49 | 55396.70 | 23802.01 | 0.89 | 0.77 | 0.53 | 0.78 | 0.86 | 0.85 | 0.86 |
| SimGNN | 2.67 | 74.70 | 973.74 | 10063.38 | 28646.82 | 55220.71 | 23681.08 | 0.86 | 0.75 | 0.55 | 0.72 | 0.84 | 0.80 | 0.82 |
| GraphSim | 3.14 | 77.85 | 991.73 | 10043.54 | 28638.96 | 55195.11 | 23674.50 | 0.84 | 0.71 | 0.51 | 0.74 | 0.81 | 0.80 | 0.79 |
| GREED | 1.87 | 77.52 | 991.55 | 10092.37 | 28715.97 | 55306.64 | 23741.21 | 0.88 | 0.77 | 0.53 | 0.77 | 0.84 | 0.84 | 0.84 |
| GEDGNN | 31.12 | 161.77 | 1181.59 | 10722.40 | 29789.05 | 56774.75 | 24727.93 | -0.03 | -0.13 | -0.25 | 0.07 | -0.06 | 0.00 | -0.06 |
| ISONET | 0.88 | 75.49 | 999.90 | 10103.25 | 28738.95 | 55336.78 | 23764.77 | 0.92 | 0.79 | 0.54 | 0.79 | 0.89 | 0.87 | 0.87 |
| H2MN | 6.20 | 83.92 | 983.70 | 10123.29 | 28773.79 | 55375.59 | 23782.46 | 0.75 | 0.67 | 0.52 | 0.60 | 0.73 | 0.69 | 0.70 |
| EGSC | 4.16 | 94.54 | 1058.17 | 10190.29 | 28937.83 | 55582.93 | 23941.03 | 0.81 | 0.70 | 0.46 | 0.76 | 0.77 | 0.79 | 0.77 |
| ERIC | 1.36 | 84.53 | 1015.90 | 10156.43 | 28836.03 | 55467.78 | 23851.80 | 0.89 | 0.75 | 0.53 | 0.78 | 0.86 | 0.85 | 0.84 |
| GRAPHEDX | 0.76 | 76.90 | 2434.89 | 2102.35 | 688.71 | 884.77 | 676.29 | 0.92 | 0.78 | 0.52 | 0.74 | 0.89 | 0.87 | 0.83 |
| **Cross-train-test-pairs** | | | | | | | | | | | | | | |
| GMN-Match | 1.65 | 64.74 | 826.78 | 10905.94 | 31729.34 | 60466.93 | 25485.93 | 0.87 | 0.74 | 0.48 | 0.76 | 0.83 | 0.83 | 0.83 |
| GMN-Embed | 1.70 | 65.99 | 830.61 | 10913.60 | 31743.89 | 60486.25 | 25499.79 | 0.88 | 0.74 | 0.47 | 0.77 | 0.83 | 0.84 | 0.83 |
| SimGNN | 2.33 | 64.49 | 818.69 | 10896.46 | 31699.60 | 60432.36 | 25462.12 | 0.82 | 0.69 | 0.51 | 0.68 | 0.80 | 0.76 | 0.78 |
| GraphSim | 2.53 | 63.70 | 819.95 | 10879.82 | 31679.72 | 60399.93 | 25442.55 | 0.84 | 0.70 | 0.49 | 0.72 | 0.81 | 0.78 | 0.79 |
| GREED | 2.30 | 66.16 | 826.86 | 10910.05 | 31730.69 | 60471.36 | 25489.28 | 0.85 | 0.72 | 0.47 | 0.74 | 0.80 | 0.81 | 0.81 |
| GEDGNN | 27.64 | 135.62 | 1000.20 | 11566.66 | 32873.11 | 62026.84 | 26511.14 | 0.06 | 0.10 | -0.11 | 0.12 | 0.04 | 0.08 | 0.07 |
| ISONET | 1.13 | 62.02 | 824.30 | 10893.56 | 31708.36 | 60438.07 | 25467.50 | 0.90 | 0.76 | 0.51 | 0.77 | 0.87 | 0.85 | 0.85 |
| H2MN | 5.05 | 69.55 | 823.61 | 10957.71 | 31799.76 | 60546.70 | 25544.55 | 0.72 | 0.62 | 0.54 | 0.58 | 0.73 | 0.65 | 0.68 |
| EGSC | 3.20 | 69.84 | 856.47 | 10946.45 | 31835.95 | 60593.43 | 25574.35 | 0.80 | 0.69 | 0.44 | 0.72 | 0.75 | 0.77 | 0.77 |
| ERIC | 1.59 | 68.33 | 839.59 | 10964.08 | 31832.56 | 60608.17 | 25577.33 | 0.89 | 0.75 | 0.49 | 0.77 | 0.84 | 0.84 | 0.85 |
| GRAPHEDX | 1.16 | 62.20 | 2685.40 | 2346.98 | 860.40 | 1252.56 | 961.71 | 0.91 | 0.75 | 0.48 | 0.72 | 0.87 | 0.84 | 0.81 |

**(1) Ktau variation remains high, confirming limited generalization.** Most models continue to exhibit significant Ktau fluctuations (up to 0.4), particularly under Cross-train-test-pairs, where performance rankings shift more drastically. This reinforces the observation that models struggle to generalize to unseen cost settings.

**(2) Node-centric models degrade with increasing edge edit costs.** As seen in the main results, models focused solely on node alignment achieve lower MSE on $\mathcal{C}_2$ but deteriorate on $\mathcal{C}_3$ and beyond. This trend extends to $\mathcal{C}_5$ and $\mathcal{C}_6$, with Cross-train-test-pairs further amplifying performance drops.

**(3) GRAPHEDX maintains its edge-aware advantage.** GRAPHEDX continues to achieve the lowest MSE on high edge-cost settings ($\mathcal{C}_3$–$\mathcal{C}_6$), demonstrating its robustness. Its relative advantage is even more pronounced in Cross-train-test-pairs, where other models struggle with increased structural complexity.

**Further Evaluation on Mutag(Table 7), Molpcba (Table 8) and Molhiv (Table 9):**

*Table 7.* Performance comparison in terms of MSE and Ktau evaluated on **Mutag** dataset for the models trained with $\mathcal{C}_0$ cost setting. Evaluation is done for different test cost configurations $\mathcal{C}_0$–$\mathcal{C}_6$ across both Intra-test-pairs and Cross-train-test-pairs test schemes.

| | MSE (Lower is Better) | | | | | | | Ktau (Higher is Better) | | | | | | |
|---|---|---|---|---|---|---|---|---|---|---|---|---|---|---|
| | $\mathcal{C}_0$ | $\mathcal{C}_1$ | $\mathcal{C}_2$ | $\mathcal{C}_3$ | $\mathcal{C}_4$ | $\mathcal{C}_5$ | $\mathcal{C}_6$ | $\mathcal{C}_0$ | $\mathcal{C}_1$ | $\mathcal{C}_2$ | $\mathcal{C}_3$ | $\mathcal{C}_4$ | $\mathcal{C}_5$ | $\mathcal{C}_6$ |
| Intra-test-pairs | | | | | | | | | | | | | | |
| GMN-Match | 1.09 | 147.23 | 3552.41 | 7355.44 | 44433.55 | 65746.84 | 32110.52 | 0.90 | 0.64 | 0.73 | 0.75 | 0.87 | 0.87 | 0.84 |
| GMN-Match | 1.09 | 147.23 | 3552.41 | 7355.44 | 44433.55 | 65746.84 | 32110.52 | 0.90 | 0.64 | 0.73 | 0.75 | 0.87 | 0.87 | 0.84 |
| GMN-Embed | 1.33 | 150.41 | 3566.08 | 7386.11 | 44497.02 | 65828.88 | 32166.08 | 0.88 | 0.63 | 0.73 | 0.73 | 0.86 | 0.85 | 0.83 |
| SimGNN | 1.74 | 149.10 | 3553.96 | 7371.02 | 44454.13 | 65778.70 | 32129.74 | 0.87 | 0.63 | 0.76 | 0.70 | 0.86 | 0.83 | 0.82 |
| GraphSim | 2.29 | 151.74 | 3573.19 | 7411.50 | 44548.73 | 65894.68 | 32209.94 | 0.84 | 0.62 | 0.75 | 0.68 | 0.83 | 0.81 | 0.80 |
| GREED | 1.70 | 151.43 | 3574.01 | 7389.28 | 44514.65 | 65846.64 | 32179.42 | 0.87 | 0.62 | 0.71 | 0.74 | 0.84 | 0.84 | 0.82 |
| GEDGNN | 33.43 | 242.82 | 4029.43 | 7795.59 | 45756.35 | 67255.40 | 33206.38 | 0.07 | -0.07 | -0.03 | 0.09 | 0.07 | 0.08 | 0.07 |
| ISONET | 1.47 | 147.04 | 3531.05 | 7318.54 | 44348.23 | 65639.56 | 32037.63 | 0.88 | 0.65 | 0.72 | 0.74 | 0.86 | 0.85 | 0.83 |
| H2MN | 1.52 | 148.69 | 3554.79 | 7368.89 | 44454.05 | 65776.56 | 32129.28 | 0.88 | 0.63 | 0.76 | 0.70 | 0.86 | 0.84 | 0.83 |
| EGSC | 1.06 | 147.83 | 3552.22 | 7357.84 | 44433.77 | 65750.16 | 32110.74 | 0.90 | 0.63 | 0.75 | 0.73 | 0.89 | 0.86 | 0.85 |
| ERIC | 1.01 | 144.92 | 3536.23 | 7340.26 | 44385.89 | 65692.41 | 32071.09 | 0.91 | 0.64 | 0.75 | 0.74 | 0.89 | 0.87 | 0.85 |
| GRAPHEDX | 1.89 | 136.90 | 405.35 | 530.75 | 524.15 | 1079.13 | 768.20 | 0.89 | 0.62 | 0.79 | 0.72 | 0.88 | 0.85 | 0.83 |
| Cross-train-test-pairs | | | | | | | | | | | | | | |
| GMN-Match | 0.96 | 120.37 | 3150.64 | 7158.31 | 42866.10 | 63434.40 | 30592.84 | 0.89 | 0.61 | 0.72 | 0.73 | 0.86 | 0.86 | 0.83 |
| GMN-Embed | 1.15 | 122.82 | 3159.48 | 7178.06 | 42905.04 | 63486.11 | 30626.90 | 0.88 | 0.61 | 0.71 | 0.72 | 0.86 | 0.85 | 0.82 |
| SimGNN | 1.42 | 121.68 | 3149.79 | 7169.85 | 42877.27 | 63455.12 | 30603.27 | 0.86 | 0.60 | 0.75 | 0.67 | 0.85 | 0.82 | 0.82 |
| GraphSim | 1.68 | 121.96 | 3162.34 | 7201.92 | 42955.05 | 63548.77 | 30667.53 | 0.85 | 0.61 | 0.75 | 0.67 | 0.83 | 0.81 | 0.80 |
| GREED | 1.47 | 124.39 | 3171.05 | 7190.65 | 42943.41 | 63528.51 | 30660.77 | 0.87 | 0.60 | 0.70 | 0.72 | 0.84 | 0.84 | 0.81 |
| GEDGNN | 27.27 | 187.86 | 3540.19 | 7512.78 | 43990.73 | 64692.51 | 31512.27 | 0.21 | 0.12 | 0.17 | 0.19 | 0.21 | 0.20 | 0.20 |
| ISONET | 1.15 | 120.75 | 3141.20 | 7139.34 | 42818.36 | 63376.75 | 30553.59 | 0.88 | 0.62 | 0.71 | 0.71 | 0.86 | 0.85 | 0.82 |
| H2MN | 1.30 | 121.38 | 3150.67 | 7169.99 | 42880.75 | 63457.65 | 30606.50 | 0.87 | 0.61 | 0.75 | 0.68 | 0.85 | 0.83 | 0.82 |
| EGSC | 0.89 | 121.92 | 3154.03 | 7166.44 | 42878.27 | 63453.89 | 30602.86 | 0.90 | 0.61 | 0.74 | 0.71 | 0.88 | 0.86 | 0.84 |
| ERIC | 0.86 | 119.94 | 3141.35 | 7149.23 | 42833.42 | 63399.50 | 30565.89 | 0.90 | 0.61 | 0.74 | 0.72 | 0.88 | 0.86 | 0.84 |
| GRAPHEDX | 1.94 | 111.23 | 378.81 | 517.51 | 542.76 | 1101.64 | 668.83 | 0.87 | 0.59 | 0.77 | 0.70 | 0.86 | 0.83 | 0.82 |

*Table 8.* Performance comparison in terms of MSE and Ktau evaluated on **Molpcba** dataset for the models trained with $\mathcal{C}_0$ cost setting. Evaluation is done for different test cost configurations $\mathcal{C}_0$–$\mathcal{C}_6$ across both Intra-test-pairs and Cross-train-test-pairs test schemes.

| | MSE (Lower is Better) | | | | | | | Ktau (Higher is Better) | | | | | | |
|---|---|---|---|---|---|---|---|---|---|---|---|---|---|---|
| | $\mathcal{C}_0$ | $\mathcal{C}_1$ | $\mathcal{C}_2$ | $\mathcal{C}_3$ | $\mathcal{C}_4$ | $\mathcal{C}_5$ | $\mathcal{C}_6$ | $\mathcal{C}_0$ | $\mathcal{C}_1$ | $\mathcal{C}_2$ | $\mathcal{C}_3$ | $\mathcal{C}_4$ | $\mathcal{C}_5$ | $\mathcal{C}_6$ |
| Intra-test-pairs | | | | | | | | | | | | | | |
| GMN-Match | 1.66 | 83.44 | 1416.16 | 5621.57 | 23515.56 | 39605.57 | 17963.42 | 0.79 | 0.61 | 0.73 | 0.62 | 0.77 | 0.74 | 0.73 |
| GMN-Embed | 2.55 | 84.61 | 1415.23 | 5639.15 | 23538.27 | 39639.93 | 17984.97 | 0.72 | 0.55 | 0.66 | 0.57 | 0.70 | 0.67 | 0.67 |
| SimGNN | 2.05 | 83.72 | 1409.27 | 5628.22 | 23511.03 | 39608.05 | 17961.36 | 0.77 | 0.60 | 0.80 | 0.56 | 0.77 | 0.70 | 0.72 |
| GraphSim | 2.25 | 84.55 | 1410.90 | 5627.74 | 23510.05 | 39606.89 | 17961.06 | 0.75 | 0.57 | 0.77 | 0.54 | 0.75 | 0.68 | 0.70 |
| GREED | 2.14 | 83.78 | 1413.23 | 5623.89 | 23512.41 | 39604.84 | 17961.52 | 0.75 | 0.58 | 0.69 | 0.60 | 0.73 | 0.70 | 0.70 |
| GEDGNN | 12.89 | 122.90 | 1577.33 | 5709.73 | 23829.84 | 39952.34 | 18227.77 | -0.03 | -0.19 | -0.12 | -0.03 | -0.03 | -0.02 | -0.07 |
| ISONET | 1.68 | 81.63 | 1408.43 | 5602.93 | 23476.72 | 39555.18 | 17930.12 | 0.79 | 0.63 | 0.74 | 0.61 | 0.77 | 0.73 | 0.73 |
| H2MN | 1.99 | 83.77 | 1409.03 | 5624.40 | 23503.96 | 39598.76 | 17955.36 | 0.78 | 0.59 | 0.80 | 0.56 | 0.78 | 0.70 | 0.72 |
| EGSC | 1.54 | 84.08 | 1415.75 | 5619.42 | 23507.63 | 39597.77 | 17957.19 | 0.80 | 0.61 | 0.77 | 0.60 | 0.80 | 0.74 | 0.75 |
| ERIC | 1.45 | 82.35 | 1409.59 | 5609.41 | 23486.78 | 39570.44 | 17938.73 | 0.81 | 0.62 | 0.78 | 0.62 | 0.80 | 0.75 | 0.76 |
| GRAPHEDX | 1.92 | 77.73 | 497.85 | 572.06 | 576.58 | 1111.81 | 720.45 | 0.78 | 0.60 | 0.67 | 0.57 | 0.78 | 0.72 | 0.69 |
| Cross-train-test-pairs | | | | | | | | | | | | | | |
| GMN-Match | 1.56 | 87.43 | 1663.68 | 5928.01 | 26752.43 | 43588.98 | 20256.75 | 0.81 | 0.60 | 0.73 | 0.64 | 0.79 | 0.76 | 0.75 |
| GMN-Embed | 2.21 | 87.83 | 1658.95 | 5936.10 | 26754.88 | 43598.13 | 20260.26 | 0.75 | 0.56 | 0.68 | 0.61 | 0.73 | 0.71 | 0.70 |
| SimGNN | 1.75 | 87.73 | 1658.74 | 5936.42 | 26754.40 | 43598.49 | 20260.33 | 0.79 | 0.59 | 0.80 | 0.58 | 0.79 | 0.71 | 0.74 |
| GraphSim | 1.80 | 87.77 | 1657.28 | 5932.05 | 26744.05 | 43586.07 | 20251.41 | 0.78 | 0.58 | 0.78 | 0.58 | 0.78 | 0.71 | 0.73 |
| GREED | 1.95 | 88.77 | 1664.97 | 5936.04 | 26762.37 | 43604.85 | 20266.61 | 0.78 | 0.57 | 0.70 | 0.62 | 0.76 | 0.73 | 0.72 |
| GEDGNN | 15.55 | 137.59 | 1891.14 | 6104.81 | 27304.82 | 44214.08 | 20723.63 | 0.07 | -0.12 | -0.01 | 0.05 | 0.08 | 0.08 | 0.04 |
| ISONET | 1.55 | 86.30 | 1658.77 | 5915.67 | 26728.29 | 43556.65 | 20236.07 | 0.80 | 0.62 | 0.74 | 0.63 | 0.79 | 0.75 | 0.75 |
| H2MN | 1.71 | 87.92 | 1658.82 | 5932.64 | 26747.47 | 43589.35 | 20254.64 | 0.79 | 0.59 | 0.80 | 0.58 | 0.79 | 0.72 | 0.74 |
| EGSC | 1.43 | 88.73 | 1666.68 | 5931.62 | 26757.85 | 43597.68 | 20262.41 | 0.82 | 0.60 | 0.77 | 0.63 | 0.82 | 0.76 | 0.77 |
| ERIC | 1.35 | 86.71 | 1659.38 | 5918.62 | 26731.26 | 43562.45 | 20239.29 | 0.83 | 0.61 | 0.77 | 0.64 | 0.82 | 0.77 | 0.77 |
| GRAPHEDX | 1.87 | 81.45 | 564.21 | 618.57 | 555.84 | 1117.07 | 754.80 | 0.80 | 0.59 | 0.68 | 0.59 | 0.80 | 0.74 | 0.71 |

*Table 9.* Performance comparison in terms of MSE and Ktau evaluated on **Molhiv** dataset for the models trained with $\mathcal{C}_0$ cost setting. Evaluation is done for different test cost configurations $\mathcal{C}_0$–$\mathcal{C}_6$ across both Intra-test-pairs and Cross-train-test-pairs test schemes.

| | MSE (Lower is Better) | | | | | | | Ktau (Lower is Better) | | | | | | |
|---|---|---|---|---|---|---|---|---|---|---|---|---|---|---|
| | $\mathcal{C}_0$ | $\mathcal{C}_1$ | $\mathcal{C}_2$ | $\mathcal{C}_3$ | $\mathcal{C}_4$ | $\mathcal{C}_5$ | $\mathcal{C}_6$ | $\mathcal{C}_0$ | $\mathcal{C}_1$ | $\mathcal{C}_2$ | $\mathcal{C}_3$ | $\mathcal{C}_4$ | $\mathcal{C}_5$ | $\mathcal{C}_6$ |
| | | | | | | | **Intra-test-pairs** | | | | | | | |
| GMN-Match | 1.50 | 160.84 | 3840.29 | 8416.02 | 50617.65 | 75265.57 | 36275.56 | 0.89 | 0.64 | 0.77 | 0.76 | 0.86 | 0.85 | 0.83 |
| GMN-Embed | 2.04 | 161.71 | 3843.86 | 8440.53 | 50661.64 | 75324.68 | 36313.30 | 0.85 | 0.62 | 0.76 | 0.74 | 0.83 | 0.82 | 0.80 |
| SimGNN | 1.80 | 164.77 | 3862.70 | 8456.25 | 50711.51 | 75381.95 | 36355.53 | 0.87 | 0.63 | 0.81 | 0.72 | 0.87 | 0.83 | 0.82 |
| GraphSim | 2.77 | 161.91 | 3857.40 | 8440.31 | 50691.16 | 75346.55 | 36336.37 | 0.85 | 0.62 | 0.80 | 0.71 | 0.84 | 0.82 | 0.79 |
| GREED | 1.90 | 161.79 | 3845.39 | 8436.04 | 50656.95 | 75316.74 | 36309.05 | 0.86 | 0.62 | 0.76 | 0.75 | 0.84 | 0.83 | 0.80 |
| GEDGNN | 39.26 | 276.78 | 4435.89 | 9007.02 | 52349.85 | 77248.83 | 37724.57 | 0.07 | -0.07 | -0.02 | 0.08 | 0.07 | 0.08 | 0.04 |
| ISONET | 1.54 | 159.22 | 3836.35 | 8413.44 | 50620.60 | 75264.45 | 36274.04 | 0.88 | 0.65 | 0.78 | 0.75 | 0.86 | 0.84 | 0.82 |
| H2MN | 1.71 | 161.87 | 3851.87 | 8443.64 | 50682.22 | 75344.95 | 36329.14 | 0.88 | 0.63 | 0.81 | 0.73 | 0.87 | 0.84 | 0.82 |
| EGSC | 1.33 | 159.67 | 3854.00 | 8435.24 | 50683.26 | 75337.21 | 36326.62 | 0.89 | 0.64 | 0.80 | 0.76 | 0.87 | 0.86 | 0.83 |
| ERIC | 1.36 | 161.19 | 3856.33 | 8433.69 | 50678.25 | 75332.26 | 36323.79 | 0.89 | 0.64 | 0.80 | 0.76 | 0.88 | 0.86 | 0.83 |
| GRAPHEDX | 2.33 | 143.36 | 677.17 | 683.42 | 679.07 | 1427.10 | 2162.89 | 0.87 | 0.63 | 0.73 | 0.72 | 0.86 | 0.82 | 0.72 |
| | | | | | | | **Cross-train-test-pairs** | | | | | | | |
| GMN-Match | 1.34 | 157.79 | 3667.22 | 8169.69 | 48121.14 | 71728.90 | 35230.53 | 0.89 | 0.64 | 0.77 | 0.76 | 0.86 | 0.85 | 0.82 |
| GMN-Embed | 1.75 | 160.00 | 3672.83 | 8185.82 | 48148.96 | 71767.77 | 35255.29 | 0.86 | 0.62 | 0.75 | 0.74 | 0.84 | 0.83 | 0.81 |
| SimGNN | 1.60 | 160.42 | 3680.07 | 8202.22 | 48188.64 | 71815.73 | 35289.07 | 0.87 | 0.63 | 0.81 | 0.72 | 0.86 | 0.83 | 0.81 |
| GraphSim | 1.97 | 158.31 | 3674.35 | 8187.67 | 48164.33 | 71780.28 | 35266.87 | 0.86 | 0.63 | 0.80 | 0.71 | 0.85 | 0.82 | 0.80 |
| GREED | 1.78 | 157.94 | 3666.98 | 8179.48 | 48136.45 | 71750.80 | 35242.79 | 0.86 | 0.63 | 0.75 | 0.74 | 0.83 | 0.83 | 0.80 |
| GEDGNN | 32.07 | 251.79 | 4161.82 | 8649.22 | 49551.41 | 73357.95 | 36424.41 | 0.17 | 0.05 | 0.09 | 0.18 | 0.17 | 0.18 | 0.15 |
| ISONET | 1.40 | 156.78 | 3667.87 | 8169.73 | 48127.16 | 71733.99 | 35232.98 | 0.88 | 0.65 | 0.78 | 0.74 | 0.86 | 0.84 | 0.82 |
| H2MN | 1.55 | 159.11 | 3674.29 | 8194.25 | 48169.90 | 71792.08 | 35273.12 | 0.87 | 0.64 | 0.81 | 0.72 | 0.86 | 0.83 | 0.81 |
| EGSC | 1.16 | 156.27 | 3672.59 | 8177.78 | 48152.66 | 71762.20 | 35253.66 | 0.90 | 0.65 | 0.80 | 0.76 | 0.88 | 0.86 | 0.83 |
| ERIC | 1.17 | 155.84 | 3666.94 | 8167.79 | 48126.55 | 71731.35 | 35231.75 | 0.90 | 0.65 | 0.80 | 0.75 | 0.88 | 0.86 | 0.83 |
| GRAPHEDX | 2.62 | 141.65 | 620.38 | 646.53 | 748.36 | 1547.26 | 2123.42 | 0.86 | 0.61 | 0.72 | 0.71 | 0.85 | 0.82 | 0.70 |

**Further Evaluation on Mutag(Table 7), Molpcba (Table 8) and Molhiv (Table 9):**

We observe **(1)** Most models continue to exhibit considerable Ktau fluctuations across cost settings, confirming their sensitivity to changes in edit cost magnitudes. **(2)** We keep observing the gap between node-centric models and edge-aware models. Notably, under Cross-train-test-pairs, these models experience even larger MSE increases as edge costs rise, suggesting that their reliance on node-level correspondences is insufficient for robust performance under varied cost settings. **(3)** The distinct advantage of GRAPHEDX in explicitly aligning both nodes and edges is further reinforced in the new datasets. **(4)** ERIC emerges as a strong performer in terms of ranking stability.

Overall, these extended results reinforce the conclusions drawn in the main paper, while providing additional evidence that models designed to explicitly align both nodes and edges concurrently, such as GRAPHEDX, consistently outperform purely node-centric models under diverse and unseen cost configurations. The added evaluation under Cross-train-test-pairs further underscores the limitations of existing approaches in generalizing beyond fixed-cost training conditions.

### A.2.3. NOTE ON UNEQUAL COST COMPATIBILITY OF NEURAL GED BASELINES

GRAPHEDX provides baseline implementation heuristics to support variable edit costs by encoding them as node features. Following this approach, we evaluate additional baselines trained under the **unequal cost setting** $C_1$, where each node's feature vector is represented as $[c_{NA}, c_{ND}, c_{EA}, c_{ED}]$. We extend our evaluation to four datasets—Mutag, Code2, Molhiv, and Molpcba—to assess how well these modified baselines adapt to variable cost configurations.

*Table 10.* Performance comparison in terms of MSE and Ktau evaluated on **Code2** dataset for the models trained with $C_1$ cost setting. Evaluation is done for different test cost configurations $C_0$–$C_6$ across both Intra-test-pairs and Cross-train-test-pairs test schemes.

| | MSE (Lower is Better) | | | | | | | Ktau (Higher is Better) | | | | | | |
|---|---|---|---|---|---|---|---|---|---|---|---|---|---|---|
| | $C_0$ | $C_1$ | $C_2$ | $C_3$ | $C_4$ | $C_5$ | $C_6$ | $C_0$ | $C_1$ | $C_2$ | $C_3$ | $C_4$ | $C_5$ | $C_6$ |
| **Intra-test-pairs** | | | | | | | | | | | | | | |
| GMN-Match | 17.80 | 13.47 | 859.04 | 9874.35 | 26404.78 | 52663.12 | 21566.95 | 0.68 | 0.78 | 0.43 | 0.49 | 0.61 | 0.70 | 0.63 |
| GMN-Embed | 27.14 | 13.42 | 911.53 | 10815.33 | 29589.43 | 53640.33 | 24616.71 | 0.69 | 0.79 | 0.71 | 0.37 | 0.55 | 0.68 | 0.44 |
| SimGNN | 79.96 | 5.21 | 756.14 | 9047.13 | 26420.19 | 51943.14 | 21828.66 | 0.16 | 0.87 | 0.16 | 0.07 | -0.26 | -0.14 | 0.05 |
| GraphSim | 54.06 | 7.40 | 4862.97 | 6119.92 | 39720.85 | 116362.43 | 88244.02 | 0.77 | 0.84 | 0.15 | 0.54 | 0.35 | 0.62 | 0.14 |
| GREED | 34.89 | 11.10 | 12339.77 | 4633.00 | 3456.97 | 10647.98 | 6819.41 | 0.79 | 0.81 | 0.59 | 0.56 | 0.75 | 0.74 | 0.72 |
| GEDGNN | 55.07 | 107.62 | 986.25 | 9568.23 | 28112.90 | 54433.86 | 23205.05 | 0.11 | 0.06 | -0.10 | 0.14 | 0.08 | 0.13 | 0.10 |
| ISONET | 157.62 | 3.02 | 1067.57 | 5806.26 | 16525.08 | 18782.09 | 8600.44 | 0.80 | 0.91 | 0.22 | 0.56 | 0.55 | 0.08 | 0.05 |
| H2MN | 56.63 | 9.00 | 2399.54 | 6989.53 | 30622.63 | 37601.68 | 30523.50 | 0.34 | 0.82 | 0.62 | -0.02 | 0.42 | -0.19 | 0.16 |
| EGSC | 98.87 | 3.96 | 1361.69 | 10726.06 | 32592.27 | 59011.18 | 27276.95 | 0.41 | 0.89 | -0.02 | 0.12 | nan | 0.26 | 0.16 |
| ERIC | 150.71 | 12.77 | 774.45 | 4369.22 | 33678.68 | 62664.45 | 19363.80 | 0.44 | 0.80 | -0.14 | 0.43 | 0.35 | 0.35 | -0.10 |
| GRAPHEDX | 1.87 | 2.25 | 1215.88 | 2025.81 | 267.99 | 2042.30 | 1003.76 | 0.90 | 0.92 | 0.72 | 0.79 | 0.90 | 0.84 | 0.83 |
| **Cross-train-test-pairs** | | | | | | | | | | | | | | |
| GMN-Match | 23.11 | 25.25 | 620.69 | 10486.17 | 28974.37 | 58187.02 | 23011.78 | 0.63 | 0.69 | 0.40 | 0.44 | 0.56 | 0.62 | 0.62 |
| GMN-Embed | 23.88 | 23.90 | 806.60 | 11731.68 | 32603.78 | 58737.56 | 26425.22 | 0.64 | 0.69 | 0.63 | 0.31 | 0.50 | 0.63 | 0.36 |
| SimGNN | 86.50 | 4.85 | 742.40 | 9750.34 | 29199.74 | 56823.73 | 23365.93 | 0.14 | 0.83 | -0.08 | 0.07 | -0.21 | -0.17 | 0.02 |
| GraphSim | 61.96 | 5.35 | 3172.15 | 5867.42 | 25800.75 | 124570.66 | 52916.74 | 0.74 | 0.83 | 0.30 | 0.54 | 0.47 | 0.62 | 0.22 |
| GREED | 38.67 | 24.66 | 11234.89 | 5445.47 | 4208.88 | 13337.61 | 8015.79 | 0.76 | 0.71 | 0.54 | 0.50 | 0.69 | 0.68 | 0.63 |
| GEDGNN | 46.09 | 83.29 | 834.90 | 10302.09 | 31086.48 | 59508.24 | 24897.12 | 0.05 | 0.09 | -0.16 | 0.09 | 0.01 | 0.08 | 0.06 |
| ISONET | 123.47 | 2.73 | 861.57 | 6539.55 | 18220.10 | 17869.69 | 7082.28 | 0.74 | 0.88 | 0.25 | 0.53 | 0.50 | 0.14 | 0.11 |
| H2MN | 54.01 | 7.00 | 2271.98 | 7504.45 | 34223.73 | 39818.31 | 32748.36 | 0.25 | 0.77 | 0.65 | -0.00 | 0.34 | -0.03 | 0.18 |
| EGSC | 102.32 | 3.33 | 1172.57 | 12154.61 | 35912.03 | 63975.67 | 27899.37 | 0.47 | 0.87 | -0.01 | 0.10 | 0.19 | 0.19 | 0.17 |
| ERIC | 147.83 | 6.66 | 667.13 | 4905.47 | 37093.44 | 68434.06 | 20866.48 | 0.17 | 0.82 | -0.29 | 0.34 | 0.27 | 0.20 | -0.27 |
| GRAPHEDX | 2.66 | 2.64 | 1263.23 | 2342.56 | 348.81 | 2774.08 | 962.25 | 0.88 | 0.89 | 0.65 | 0.76 | 0.88 | 0.81 | 0.81 |

*Table 11.* Performance comparison in terms of MSE and Ktau evaluated on **Mutag** dataset for the models trained with $C_1$ cost setting. Evaluation is done for different test cost configurations $C_0$–$C_6$ across both Intra-test-pairs and Cross-train-test-pairs test schemes.

| | MSE (Lower is Better) | | | | | | | Ktau (Higher is Better) | | | | | | |
|---|---|---|---|---|---|---|---|---|---|---|---|---|---|---|
| | $C_0$ | $C_1$ | $C_2$ | $C_3$ | $C_4$ | $C_5$ | $C_6$ | $C_0$ | $C_1$ | $C_2$ | $C_3$ | $C_4$ | $C_5$ | $C_6$ |
| **Intra-test-pairs** | | | | | | | | | | | | | | |
| GMN-Match | 88.56 | 69.72 | 1688.50 | 5276.92 | 25896.03 | 40496.25 | 19455.97 | 0.83 | 0.60 | 0.54 | 0.43 | 0.57 | 0.48 | 0.53 |
| GMN-Embed | 89.73 | 73.08 | 3707.93 | 6151.83 | 44783.81 | 60048.02 | 33096.20 | 0.81 | 0.60 | 0.56 | 0.58 | 0.64 | 0.61 | 0.39 |
| SimGNN | 30.82 | 5.18 | 2591.59 | 5465.69 | 40103.21 | 60833.75 | 30325.59 | 0.42 | 0.86 | 0.67 | -0.10 | -0.23 | -0.12 | -0.18 |
| GraphSim | 42.43 | 5.82 | 20210.07 | 5378.95 | 274879.94 | 694983.25 | 153201.47 | 0.55 | 0.85 | 0.78 | 0.53 | 0.61 | 0.59 | 0.62 |
| GREED | 8.10 | 69.24 | 297.66 | 1780.65 | 6527.98 | 8541.12 | 4573.95 | 0.76 | 0.61 | 0.75 | 0.57 | 0.74 | 0.71 | 0.69 |
| GEDGNN | 87.87 | 152.46 | 3253.90 | 6491.89 | 42534.49 | 63257.59 | 30456.29 | 0.48 | 0.44 | 0.45 | 0.45 | 0.46 | 0.47 | 0.46 |
| ISONET | 126.40 | 3.83 | 2116.40 | 4114.26 | 26038.34 | 33789.64 | 18345.88 | 0.65 | 0.88 | 0.56 | 0.40 | 0.06 | 0.12 | 0.04 |
| H2MN | 173.99 | 3.87 | 1275.88 | 2033.74 | 5760.80 | 8705.74 | 4497.13 | 0.58 | 0.87 | 0.64 | 0.45 | 0.54 | 0.51 | 0.52 |
| EGSC | 334.60 | 2.25 | 3151.06 | 6411.25 | 45837.08 | 67383.50 | 33178.26 | 0.48 | 0.91 | 0.24 | 0.35 | -0.16 | -0.15 | -0.12 |
| ERIC | 35.63 | 72.53 | 3738.90 | 7405.44 | 44886.48 | 66182.52 | 32528.36 | 0.64 | 0.62 | 0.04 | 0.32 | -0.11 | -0.35 | -0.03 |
| GRAPHEDX | 1.22 | 2.53 | 664.83 | 940.66 | 439.79 | 1229.95 | 2650.44 | 0.91 | 0.91 | 0.92 | 0.79 | 0.89 | 0.85 | 0.73 |
| **Cross-train-test-pairs** | | | | | | | | | | | | | | |
| GMN-Match | 82.96 | 64.65 | 1480.63 | 5120.59 | 24491.29 | 38170.34 | 18127.36 | 0.82 | 0.57 | 0.48 | 0.40 | 0.57 | 0.47 | 0.52 |
| GMN-Embed | 85.82 | 70.91 | 3258.56 | 6016.66 | 43038.82 | 58006.16 | 31318.47 | 0.79 | 0.56 | 0.52 | 0.53 | 0.62 | 0.59 | 0.38 |
| SimGNN | 26.78 | 4.21 | 2348.13 | 5165.00 | 38347.61 | 58458.98 | 28708.43 | 0.40 | 0.85 | 0.62 | -0.09 | -0.25 | -0.15 | -0.22 |
| GraphSim | 41.94 | 4.23 | 18256.14 | 4577.97 | 242144.36 | 617758.94 | 133718.81 | 0.52 | 0.85 | 0.76 | 0.51 | 0.59 | 0.57 | 0.61 |
| GREED | 8.80 | 65.24 | 276.88 | 2028.84 | 6333.84 | 9802.24 | 4602.12 | 0.74 | 0.58 | 0.73 | 0.51 | 0.73 | 0.66 | 0.68 |
| GEDGNN | 85.37 | 122.47 | 2837.45 | 6235.55 | 40849.52 | 60787.81 | 28847.61 | 0.46 | 0.41 | 0.42 | 0.43 | 0.43 | 0.45 | 0.43 |
| ISONET | 100.43 | 2.88 | 1994.19 | 4062.93 | 25785.42 | 32965.49 | 17438.57 | 0.61 | 0.88 | 0.51 | 0.33 | -0.02 | 0.04 | -0.00 |
| H2MN | 148.14 | 3.80 | 1219.73 | 1947.56 | 5059.34 | 7298.89 | 4007.83 | 0.55 | 0.86 | 0.64 | 0.41 | 0.53 | 0.51 | 0.48 |
| EGSC | 313.58 | 2.07 | 2835.02 | 6374.85 | 44081.77 | 64827.62 | 31500.39 | 0.43 | 0.90 | 0.24 | 0.35 | -0.17 | -0.14 | -0.13 |
| ERIC | 32.09 | 65.77 | 3280.24 | 7156.61 | 43117.18 | 63487.73 | 30849.33 | 0.62 | 0.59 | 0.05 | 0.30 | -0.15 | -0.36 | -0.07 |
| GRAPHEDX | 1.13 | 2.67 | 641.48 | 885.68 | 402.79 | 1062.02 | 2348.17 | 0.90 | 0.90 | 0.91 | 0.78 | 0.88 | 0.85 | 0.71 |

*Table 12.* Performance comparison in terms of MSE and Ktau evaluated on **Molpcba** dataset for the models trained with $\mathcal{C}_1$ cost setting. Evaluation is done for different test cost configurations $\mathcal{C}_0$–$\mathcal{C}_6$ across both Intra-test-pairs and Cross-train-test-pairs test schemes.

| | MSE (Lower is Better) | | | | | | | Ktau (Higher is Better) | | | | | | |
|---|---|---|---|---|---|---|---|---|---|---|---|---|---|---|
| | $\mathcal{C}_0$ | $\mathcal{C}_1$ | $\mathcal{C}_2$ | $\mathcal{C}_3$ | $\mathcal{C}_4$ | $\mathcal{C}_5$ | $\mathcal{C}_6$ | $\mathcal{C}_0$ | $\mathcal{C}_1$ | $\mathcal{C}_2$ | $\mathcal{C}_3$ | $\mathcal{C}_4$ | $\mathcal{C}_5$ | $\mathcal{C}_6$ |
| | *Intra-test-pairs* | | | | | | | | | | | | | |
| GMN-Match | 33.35 | 24.95 | 1094.62 | 4338.30 | 18529.18 | 24572.72 | 13495.41 | 0.68 | 0.59 | 0.25 | 0.14 | 0.11 | 0.30 | 0.09 |
| GMN-Embed | 114.43 | 29.49 | 1123.06 | 4499.16 | 13038.17 | 26686.06 | 9212.62 | 0.69 | 0.53 | 0.08 | 0.18 | 0.26 | 0.30 | 0.29 |
| SimGNN | 45.83 | 4.48 | 1164.50 | 1504.99 | 13823.54 | 26120.01 | 9697.23 | 0.55 | 0.80 | -0.29 | -0.11 | -0.25 | -0.20 | -0.23 |
| GraphSim | 18.50 | 4.93 | 24955.04 | 4324.32 | 423399.34 | 1154531.75 | 86381.91 | 0.59 | 0.78 | 0.68 | 0.50 | 0.53 | 0.53 | 0.54 |
| GREED | 317.55 | 27.04 | 765.63 | 3097.60 | 2515.16 | 9035.11 | 1888.99 | 0.57 | 0.54 | 0.53 | 0.35 | 0.66 | 0.44 | 0.57 |
| GEDGNN | 49.13 | 74.51 | 1256.97 | 4864.74 | 22103.42 | 37670.08 | 16721.63 | 0.39 | 0.35 | 0.42 | 0.30 | 0.38 | 0.36 | 0.37 |
| ISONET | 69.82 | 3.75 | 745.64 | 3802.85 | 14000.82 | 25897.02 | 12052.43 | 0.60 | 0.81 | 0.55 | 0.47 | 0.16 | 0.15 | 0.28 |
| H2MN | 630.60 | 4.41 | 857.49 | 1030.92 | 21498.60 | 37004.68 | 16198.50 | 0.38 | 0.80 | 0.78 | 0.50 | 0.59 | 0.55 | 0.61 |
| EGSC | 127971.02 | 3.14 | 57688.06 | 17323.34 | 51966.68 | 35863.64 | 58844.41 | -0.24 | 0.83 | 0.25 | 0.33 | -0.24 | -0.19 | -0.21 |
| ERIC | 56.28 | 3.07 | 1998.92 | 6666.78 | 25844.59 | 42596.17 | 19992.22 | 0.50 | 0.84 | 0.19 | 0.34 | -0.11 | -0.11 | -0.09 |
| GRAPHEDX | 2.23 | 3.06 | 748.43 | 1422.49 | 491.72 | 2124.67 | 2018.55 | 0.81 | 0.84 | 0.91 | 0.63 | 0.80 | 0.73 | 0.63 |
| | *Cross-train-test-pairs* | | | | | | | | | | | | | |
| GMN-Match | 40.75 | 32.26 | 1242.33 | 4557.17 | 21136.06 | 27562.35 | 15166.63 | 0.69 | 0.58 | 0.28 | 0.15 | 0.12 | 0.30 | 0.11 |
| GMN-Embed | 141.12 | 36.65 | 1242.18 | 4691.41 | 14494.54 | 29162.00 | 10283.75 | 0.71 | 0.53 | 0.10 | 0.21 | 0.29 | 0.31 | 0.30 |
| SimGNN | 48.81 | 3.82 | 1359.52 | 1775.02 | 16854.24 | 29971.59 | 11820.60 | 0.59 | 0.80 | -0.34 | -0.16 | -0.29 | -0.25 | -0.27 |
| GraphSim | 23.11 | 3.97 | 30216.50 | 6625.60 | 496550.09 | 1310088.12 | 127795.49 | 0.58 | 0.79 | 0.68 | 0.50 | 0.53 | 0.53 | 0.54 |
| GREED | 407.66 | 36.07 | 888.94 | 3386.05 | 2554.08 | 8852.97 | 1901.29 | 0.59 | 0.53 | 0.53 | 0.38 | 0.68 | 0.47 | 0.59 |
| GEDGNN | 47.31 | 87.41 | 1545.84 | 5227.07 | 25470.30 | 41820.07 | 19127.62 | 0.40 | 0.35 | 0.43 | 0.32 | 0.39 | 0.38 | 0.39 |
| ISONET | 74.78 | 3.34 | 912.98 | 4038.52 | 16425.68 | 29175.82 | 13859.29 | 0.59 | 0.81 | 0.54 | 0.45 | 0.15 | 0.14 | 0.31 |
| H2MN | 612.09 | 3.79 | 1044.20 | 968.53 | 24597.21 | 40843.65 | 18367.26 | 0.32 | 0.80 | 0.78 | 0.50 | 0.57 | 0.54 | 0.60 |
| EGSC | 126122.16 | 2.96 | 55322.41 | 15597.97 | 49923.90 | 34827.25 | 56480.20 | -0.29 | 0.83 | 0.21 | 0.28 | -0.28 | -0.24 | -0.25 |
| ERIC | 67.26 | 2.82 | 2292.01 | 7025.28 | 29339.78 | 46860.56 | 22501.54 | 0.48 | 0.84 | 0.15 | 0.33 | -0.16 | -0.15 | -0.13 |
| GRAPHEDX | 2.13 | 2.90 | 794.40 | 1458.75 | 496.47 | 2053.95 | 2431.26 | 0.82 | 0.84 | 0.91 | 0.65 | 0.81 | 0.75 | 0.62 |

*Table 13.* Performance comparison in terms of MSE and Ktau evaluated on **Molhiv** dataset for the models trained with $\mathcal{C}_1$ cost setting. Evaluation is done for different test cost configurations $\mathcal{C}_0$–$\mathcal{C}_6$ across both Intra-test-pairs and Cross-train-test-pairs test schemes.

| | MSE (Lower is Better) | | | | | | | Ktau (Higher is Better) | | | | | | |
|---|---|---|---|---|---|---|---|---|---|---|---|---|---|---|
| | $\mathcal{C}_0$ | $\mathcal{C}_1$ | $\mathcal{C}_2$ | $\mathcal{C}_3$ | $\mathcal{C}_4$ | $\mathcal{C}_5$ | $\mathcal{C}_6$ | $\mathcal{C}_0$ | $\mathcal{C}_1$ | $\mathcal{C}_2$ | $\mathcal{C}_3$ | $\mathcal{C}_4$ | $\mathcal{C}_5$ | $\mathcal{C}_6$ |
| | *Intra-test-pairs* | | | | | | | | | | | | | |
| GMN-Match | 84.22 | 77.21 | 4275.91 | 6678.87 | 51038.91 | 74439.42 | 35349.29 | 0.82 | 0.62 | 0.51 | 0.63 | 0.34 | 0.43 | 0.39 |
| GMN-Embed | 107.67 | 78.62 | 2927.40 | 5536.99 | 25347.75 | 44881.81 | 21171.43 | 0.79 | 0.61 | 0.61 | 0.57 | 0.36 | 0.37 | 0.55 |
| SimGNN | 1275.56 | 4.43 | 2996.78 | 5817.63 | 44547.96 | 67305.20 | 31133.68 | -0.03 | 0.87 | -0.24 | -0.09 | -0.18 | -0.15 | -0.18 |
| GraphSim | 74.41 | 6.93 | 47927.28 | 8157.99 | 896704.50 | 2437351.50 | 352252.34 | 0.60 | 0.85 | 0.78 | 0.65 | 0.71 | 0.64 | 0.66 |
| GREED | 31.87 | 78.62 | 400.23 | 1858.71 | 8493.46 | 16178.17 | 4247.03 | 0.47 | 0.60 | 0.70 | 0.59 | 0.78 | 0.65 | 0.76 |
| GEDGNN | 102.62 | 163.43 | 3487.30 | 7343.57 | 48346.82 | 72261.33 | 34322.52 | 0.42 | 0.38 | 0.45 | 0.37 | 0.41 | 0.40 | 0.40 |
| ISONET | 147.50 | 3.73 | 2228.07 | 3969.70 | 63199.26 | 215335.28 | 11747.15 | 0.65 | 0.87 | 0.50 | 0.23 | -0.07 | -0.13 | 0.23 |
| H2MN | 195.87 | 4.09 | 2536.28 | 1167.80 | 9423.36 | 10939.88 | 6988.54 | 0.58 | 0.87 | 0.71 | 0.56 | 0.81 | 0.71 | 0.78 |
| EGSC | 107.62 | 2.65 | 54097.84 | 42222.03 | 52102.21 | 77055.54 | 37587.10 | 0.76 | 0.90 | -0.24 | -0.13 | -0.08 | -0.05 | -0.07 |
| ERIC | 112.62 | 3.67 | 2891.03 | 6922.14 | 48256.91 | 72547.16 | 34088.49 | 0.59 | 0.89 | 0.76 | 0.56 | 0.47 | 0.43 | 0.46 |
| GRAPHEDX | 1.67 | 2.55 | 1144.66 | 1723.36 | 659.43 | 2116.98 | 6071.16 | 0.90 | 0.90 | 0.94 | 0.76 | 0.86 | 0.83 | 0.61 |
| | *Cross-train-test-pairs* | | | | | | | | | | | | | |
| GMN-Match | 79.76 | 70.00 | 4079.30 | 6494.20 | 48387.97 | 70760.85 | 34252.86 | 0.81 | 0.61 | 0.51 | 0.62 | 0.32 | 0.42 | 0.36 |
| GMN-Embed | 97.68 | 77.28 | 2809.92 | 5400.62 | 24487.42 | 42955.95 | 20811.11 | 0.78 | 0.59 | 0.58 | 0.57 | 0.33 | 0.35 | 0.53 |
| SimGNN | 1337.13 | 3.91 | 2716.70 | 5514.31 | 41919.62 | 63642.57 | 29892.39 | 0.04 | 0.88 | -0.17 | -0.03 | -0.13 | -0.10 | -0.11 |
| GraphSim | 68.33 | 4.30 | 47257.94 | 8083.15 | 887956.00 | 2442757.25 | 341169.41 | 0.59 | 0.87 | 0.79 | 0.65 | 0.72 | 0.65 | 0.66 |
| GREED | 31.38 | 72.56 | 453.63 | 1856.98 | 8266.98 | 15416.55 | 4394.33 | 0.46 | 0.60 | 0.67 | 0.58 | 0.78 | 0.64 | 0.75 |
| GEDGNN | 102.07 | 145.83 | 3254.97 | 7036.75 | 45692.87 | 68545.85 | 33114.59 | 0.41 | 0.40 | 0.45 | 0.35 | 0.40 | 0.39 | 0.40 |
| ISONET | 145.29 | 3.35 | 2135.23 | 3804.58 | 60669.10 | 209385.00 | 11648.02 | 0.65 | 0.88 | 0.53 | 0.23 | -0.08 | -0.09 | 0.19 |
| H2MN | 202.61 | 3.73 | 2357.89 | 1086.99 | 8995.01 | 10162.74 | 6971.52 | 0.58 | 0.88 | 0.71 | 0.56 | 0.80 | 0.70 | 0.77 |
| EGSC | 103.66 | 2.32 | 52582.15 | 40584.48 | 49878.92 | 73829.79 | 36611.28 | 0.76 | 0.91 | -0.17 | -0.07 | -0.05 | -0.02 | -0.01 |
| ERIC | 90.50 | 2.61 | 2715.11 | 6634.09 | 46164.62 | 69539.44 | 33350.18 | 0.58 | 0.90 | 0.75 | 0.55 | 0.43 | 0.39 | 0.42 |
| GRAPHEDX | 1.59 | 2.50 | 1111.27 | 1645.50 | 631.74 | 2022.22 | 5596.26 | 0.90 | 0.91 | 0.94 | 0.76 | 0.86 | 0.83 | 0.61 |

We present the results for Code2(Table 11) Mutag(Table 11), Molpcba (Table 12), Molhiv (Table 9). The key observations are as follows:

**(1)** Compared to models trained under equal-cost settings ($\mathcal{C}_0$), some models exhibit improved Ktau stability across cost variations, indicating better adaptation. However, significant Ktau fluctuations persist, with drops of up to 0.4 for certain models, showing that training with $\mathcal{C}_1$ alone is insufficient for full generalization to arbitrary cost configurations.

**(2)** GRAPHEDX achieves the lowest MSE across most cost settings, particularly in high-cost regimes ($\mathcal{C}_3$–$\mathcal{C}_6$), confirming its robustness to structural variations. It also maintains the highest Ktau values, indicating that it preserves correct rank orderings better than other baselines.

**(3)** Models such as GMN-Match, GMN-Embed, and SimGNN continue to perform well on low-cost settings ($\mathcal{C}_0$–$\mathcal{C}_2$) but degrade significantly as edge edit costs increase. The trend observed in $\mathcal{C}_3$–$\mathcal{C}_6$ suggests that training with unequal costs does not fully enable these models to capture edge transformations effectively.

**(4)** ISONET achieves competitive Ktau scores, particularly in $\mathcal{C}_1$, suggesting its predictions remain relatively well-ordered. However, its MSE remains high on high-cost settings, implying that it does not fully account for large structural changes despite its ranking stability.

**(5)** GREED performs well in terms of Ktau, particularly in $\mathcal{C}_3$–$\mathcal{C}_6$, where it ranks second after GRAPHEDX. However, GraphSim exhibits high MSE variability, performing poorly on certain cost settings, indicating limited adaptability to cost variations.

**(6)** GEDGNN continues to struggle, with near-zero or negative Ktau in multiple settings, reaffirming its lack of generalization across cost configurations.

**Conclusion.** Training with unequal cost settings ($\mathcal{C}_1$) helps improve generalization to some extent, but most models still struggle to adapt to high-cost variations, particularly those focused only on node alignment. GRAPHEDX remains the most robust model across different cost settings, both in terms of MSE and Ktau. Future work should explore more flexible architectures that explicitly integrate variable cost structures into their learning framework to improve generalization further.

