# OpenReview forum: "Position: Graph Matching Systems Deserve Better Benchmarks"
_ICML.cc/2025/Position_Paper_Track — ICML 2025 Position Paper Track poster_

### Official Review · Reviewer_9jMo · 2025-03-04

**Significance:** 3
**Argument Clarity:** 3
**Rating:** 4
**Confidence:** 1

**Questions:**

Can authors discuss more about how experimental observations affect future research directions? Or, what is the practical impact on real-world applications?

**Discussion Potential:**

3

**Paper Summary:**

This paper establishes the severity of issues in the current development of graph-matching systems, which stem primarily from the lack of reliable benchmarks. The contributions include data leakage analysis, deeper understanding of variabl cost graph edit distance (GED), and adversarial test set design.

By identifying pervasive train-test leakage, this paper shows that test graph pairs being isomorphic to training graph pairs constitutes leakage. Then, through investigating principles governing optimal edit paths and edit costs in variable cost GED, this paper instantiates generating high-quality training data at scale. Further, evaluation drawbacks have been mitigated by the adversarial test sets that evaluate whether models can identify the correct ground truth permutations.

**Position:**

Yes

**Position In Title:**

Yes

**Related Work:**

3

**Strengths And Weaknesses:**

Strengths

+ This paper gives a comprehensive investigation of current graph-matching benchmarks from a wide range of papers. Their analysis of various datasets reveals a significant number of structurally isomorphic graphs. Modern GNN-based neural GED models produce identical embeddings for graphs within the same isomorphism class. In Section 5.3, the insight observation has been summarized.

+ The extensive empirical support under varying cost settings reveals that current neural models is difficult in generalization. The experimental results explain the demand of robust architectures. The extensive experimentation with multiple cost settings strengthens the claims regarding model generalization issues.

Weakness
- The authors show the insightful observations in the experiments. It seems that no much discussion on impact of these observations. It would be better to discuss more about how these observations affect on the future research directions.

**Support:**

3

---

> ### Author Rebuttal · Authors · 2025-04-01
>
> Thank you for your thoughtful and encouraging review. We appreciate your recognition of our empirical analysis and will expand the discussion section to more clearly articulate how our findings motivate future research directions in graph matching.
>
> > Can authors discuss more about how experimental observations affect future research directions? Or, what is the practical impact on real-world applications?
>
> In the revised version, we will expand our discussion section to more clearly articulate the implications of our findings.
>
> In particular, our analysis highlights two key challenges:
> (1) the lack of generalization in current models across cost configurations, and
> (2) the inflated performance caused by train-test leakage.
>
> These observations suggest that future work should prioritize:
> - **Designing architectures that are cost-aware**, explicitly conditioning on edit cost vectors to ensure robustness under deployment-time variability.
> - **Developing evaluation protocols and benchmarks** that rigorously test generalization across structural and cost variations—moving beyond uniform-cost GED and fixed-pair datasets.
> - **Creating interpretable and alignment-consistent models**, especially for real-world domains (e.g., drug design, program synthesis), where alignment paths are crucial for downstream reasoning.
>
> We will incorporate these points in the discussion section to better connect our experimental insights with future directions and practical applications.

---

### Official Review · Reviewer_FZqU · 2025-03-10

**Significance:** 2
**Argument Clarity:** 2
**Rating:** 3
**Confidence:** 3

**Questions:**

1. Regarding lines 135-145, could you explain why this is important? I believe it shouldn't be difficult to achieve the desired behavior using the original library.

2. What is the purpose of Algorithm 1? Was it proposed in previous papers, or is it an original contribution from your team? Is it used to generate the ground truth labels for graph matching systems?

3. Do you think train-test leakage as merely a design flaw, or is it a real issue encountered in practical applications? In other words, do actual graph matching problems also exhibit instances of train-test leakage (in this sense, it's a feature but not a bug)?

**Discussion Potential:**

2

**Paper Summary:**

This position paper argues that current benchmarks used to evaluate graph matching systems, particularly those for Graph Edit Distance (GED), have significant limitations: (1) extensive train-test leakage in widely used datasets, where isomorphic graphs appear in both training and test sets; (2) poor generalization capability of current neural models when tested on truly distinct graph structures; and (3) insufficient consideration of variable edit costs in benchmark design, which underutilizes GED's flexibility. Based on empirical analysis and theoretical insights, the authors propose improved guidelines for designing and evaluating neural network models for graph matching, including proper dataset construction to prevent leakage and testing protocols that evaluate models' ability to generalize across different cost configurations.

**Position:**

Yes

**Position In Title:**

Yes

**Related Work:**

2

**Strengths And Weaknesses:**

Strengths:
1. The position is well presented in the introduction with clear logic.

Weaknesses:
1. **Discrepancy between Title and Content**: The title of this position paper mentions "Graph Matching," but the paper mainly focuses on "graph edit distance." While these two concepts are related, they remain distinct. I recommend that the authors clarify this issue, particularly in the introductory sections.

2. **Related Work**: The section on related work is presented generally, merely introducing the basic ideas of various methods. To align it with creating a "better graph matching benchmark," I suggest expanding the discussion in lines 124-137.

3. You raise the "variable-cost GED" concept in Chapter 4. However, it's not rigorously defined in previous sections.

4. The logic transition from the first three chapters to the fourth chapter is unnatural. Why do we need to analyze the variable-cost GED? It's not well motivated.

5. What's the connection between Section 5.4, 5.5, and previous subsections? There's no motivation for these experiments.

As a summary, I think there are two main weaknesses:
1, This paper doesn't "look like" a position paper. I appreciate the authors' efforts in conducting those comprehensive benchmarks, but the discussion on position and related works is not insightful.
2. There's no clear logic transition across different chapters. Authors need to motivate the value of proposed experiments.

**Support:**

2

---

> ### Author Rebuttal · Authors · 2025-04-01
>
> We thank the reviewer for their detailed feedback.
> >title mentions "Graph Matching," but the paper focuses on "graph edit distance."
>
> As noted around line 42, GED serves as a generalized framework for graph matching, capable of capturing a wide range of structural similarity notions—including subgraph matching, maximum common subgraph, and equal-cost GED—through its flexible *variable-cost* formulation. We will revise the introduction suitably to prominently clarify this notion.
> >Related Work...merely introducing the basic ideas of various methods...suggest expanding the discussion.
>
> Our emphasis is on evaluation and data preparation rather than model design. Since all  baselines employ permutation-invariant GNNs, our primary concern is how these models are trained and benchmarked—analyzed in detail in Section 5.1. If the reviewer has specific suggestions to enhance this discussion, we would be glad to incorporate them.
> >"variable-cost GED" concept in Chapter 4 ... not rigorously defined in previous sections.
>
> The definition of GED in Section 2 already supports variable cost. For clarity, we will explicitly highlight this, distinguishing it from the commonly used equal-cost setting where all edit costs are set to 1. This distinction is key to GED’s modeling flexibility.
> >logic transition...to the fourth chapter...Why…analyze the variable-cost GED?
>
> The motivation for analyzing variable-cost GED is closely tied to our goal of generating structurally diverse, leakage-free datasets. Chapter 4 highlights a key theoretical insight: optimal alignments remain invariant under a broad range of cost settings. This allows a single combinatorial solver run to generate multiple GED datasets with varied cost configurations, enabling both scalable data augmentation and cost-based stress tests for generalization. While Section 4.2 outlines these implications, we will revise the start of Chapter 4 to better foreshadow its role in addressing data sparsity and evaluation robustness, and to clarify its connection to Section 5.
> >connection between Section 5.4, 5.5, and previous subsections
>
> Building on the invariance result in Section 4, Sections 5.4 and 5.5 put into practice the two key implications outlined in Section 4.2—scalable data augmentation and adversarial testing—directly addressing limitations raised earlier in Section 5. To clarify this connection, we will revise the beginning of Section 5 to explicitly outline the role and intent of each subsection, and provide contextual framing for Sections 5.4 and 5.5.
> >doesn't "look like" a position paper
>
> We believe we have a strong position with regard to the lacunae in the training and evaluation of neural graph matching methods.  Supported by strong empirical evidence, we advocate moving from fixed-cost, static datasets to cost-invariant, alignment-centric, leakage-free evaluation.  While our extensive experimental work might have made the current version look somewhat dissimilar to typical position papers, we would appreciate further inputs on how to retain the strong experiments *and* present a clear position.  We have already taken care of a few clarifying transition paragraphs.
> >lines 135-145...why this is important
>
> To test the invariance hypothesis in Section 4, we examined existing libraries for retrieving all optimal edit paths. NetworkX returns all such paths by default, but GEDLIB—used for larger graphs—only provides one. To maintain reproducibility and avoid modifying standard solvers, we ran our tests using default settings. Even then, predicted alignments remained stable, validating our hypothesis. We'll clarify this in the revision.
> >Algorithm 1...original contribution?
>
> Algorithm 1 is an original contribution. As explained in Section 4.1, it converts a permutation matrix into a valid, interpretable edit path—highlighting their tight coupling. Crucially, it decouples the edit path from the final GED value, enabling us to formalize the invariance of edit paths under cost changes. This enables both efficient GED recomputation across cost settings and interpretable model evaluation. We will include this explanatory text at a suitable point.
> >train-test leakage: feature or bug
>
> In a real application, a learning method may be called upon to perform two kinds of inference: rote learning, and generalization within a distribution (the more common case). Since rote recall can be handled with simple lookup tables, expensive learning methods must be evaluated primarily on their ability to generalize.  Our position is that benchmarking neural graph matching models must focus on this generalization, as motivated in lines 15–27.  This concern has been echoed in recent work on structural diversity in graph learning [1,2]. We will include this in a discussion of alternative perspectives.
>
> [1] *Towards Better Out-of-Distribution Generalization of Neural Algorithmic Reasoning Tasks. TMLR 2023.*
>
> [2] *Challenges of Generating Structurally Diverse Graphs. NeurIPS 2024.*

---

> > ### Comment · Reviewer_FZqU · 2025-04-02
> >
> > Thanks for the comments. I think most of my concerns have been acknowledged. Please be sure to make corresponding changes in the camera-ready version.

---

### Official Review · Reviewer_4idq · 2025-03-13

**Significance:** 3
**Argument Clarity:** 3
**Rating:** 4
**Confidence:** 2

**Questions:**

I would like to note that I am not familiar with the graph matching literature, though I am familiar with some non-neural techniques, optimal transport, and GNNs.

**Discussion Potential:**

3

**Paper Summary:**

This paper considers benchmarks for graph matching systems that seek to approximate graph edit distance, especially neural ones. They show a major issue in existing benchmarks: there are many graphs in common benchmarks that are isomorphic to each other, even across train and test splits. This is especially relevant because permutation invariant neural graph embeddings will embed these graphs identically. They show that eliminating this leakage can significantly degrade the performance of neural graph matching methods. Furthermore, they prove a result that optimal edit paths are identical under certain cost settings, and use this to develop new data augmentation procedures to quickly compute GEDs across different cost settings.

### Update after rebuttal

The authors have replied to my rebuttal. I still think the weakness I stated holds, and I do not really see the point of the adversarial test-time cost changes. But, I overall maintain that this is a good contribution to the field, and recommend acceptance.

**Position:**

Yes

**Position In Title:**

Yes

**Related Work:**

4

**Strengths And Weaknesses:**

Strengths
1. In general, the exposition and contextualization of the background is good. The exposition and history in Section 5.1 is great.
2. Massive amounts of train-test leakage are found in terms of isomorphic graphs. The observation that permutation invariant GNNs assign the same embedding to isomorphic graphs has a very important implication in graph matching studies.
3. Theorem 4.2 and the use of it as a data augmentation is really nice (though I did not check the validity of the proofs). Overall, the method of making datasets that do not have graph-isomorphism leakage seems useful for the field.
4. Inclusion of code, data, and many experiments across many models make this a well-supported position paper.


Weaknesses
1. The point of these adversarial test-time cost changes is not so clear to me, yet many tables in the main paper and appendix are dedicated to it. It is somewhat interesting that kendall tau often does not completely degrade when perturbing cost. But that MSE is so large seems pretty obvious: the neural networks are trained for a completely different objective. This line from the Appendix is a good comment on this: "Future work should explore more flexible architectures that explicitly integrate variable cost structures into their learning framework to improve generalization further", but I'm not sure what else can be inferred.
2. An interpretable reference for the MSE and Ktau scores would be beneficial. For instance, comparing against approximate solvers would be nice, but understandably more experiments may be a lot to ask for for a position paper.

**Support:**

4

---

> ### Author Rebuttal · Authors · 2025-04-01
>
> We thank the reviewer for their thoughtful evaluation and for highlighting both the broader contributions and technical insights of the paper.
>
> >The point of these adversarial test-time cost changes is not so clear to me
>
> The motivation behind our adversarial test-time cost perturbation experiments is to evaluate whether models have learned meaningful and generalizable alignments, beyond fitting GED values under a fixed cost. Neural approximators like GraphEDX and ERIC internally estimate alignments, but current benchmarks do not assess how close these are to the true alignments.
>
> As established in Section 4, the optimal alignment remains invariant across a broad range of cost settings. Hence, if a model’s predicted alignment is accurate, its GED predictions should remain consistent under cost perturbations, yielding low MSE. Conversely, large shifts in MSE indicate poor alignment generalization. The observation that some models maintain stable rankings despite MSE degradation suggests partial alignment fidelity and highlights the diagnostic value of our evaluation.

---

> > ### Comment · Reviewer_4idq · 2025-04-03
> >
> > Thank you for the rebuttal, I maintain recommendation of acceptance.

---

### Decision · Program_Chairs · 2025-04-30

**Decision:**

Accept (poster)

**Comment:**

The reviewers are in agreement: the paper makes a notable discovery that current neural models do not generalize well due to train-test leakage. The investigation is extensive, and the authors support their position in a systematic way. During the reviewing stage, the authors interacted with the reviewers to successfully address remaining concerns. I recommend acceptance.